# Deep-LASI: deep-learning assisted, single-molecule imaging analysis of multi-color DNA origami structures

Simon Wanninger [1], Pooyeh Asadiatouei [1], Johann Bohlen [1], Clemens-Bässem Salem[1], Philip Tinnefeld [1], Evelyn Ploetz [1] ✉ & Don C. Lamb [1] ✉

Single-molecule experiments have changed the way we explore the physical world, yet data analysis remains time-consuming and prone to human bias. Here, we introduce Deep-LASI (Deep-Learning Assisted Single-molecule Imaging analysis), a software suite powered by deep neural networks to rapidly analyze single-, two- and three-color single-molecule data, especially from single-molecule Förster Resonance Energy Transfer (smFRET) experiments. Deep-LASI automatically sorts recorded traces, determines FRET correction factors and classifies the state transitions of dynamic traces all in ~20–100 ms per trajectory. We benchmarked Deep-LASI using ground truth simulations as well as experimental data analyzed manually by an expert user and compared the results with a conventional Hidden Markov Model analysis. We illustrate the capabilities of the technique using a highly tunable L-shaped DNA origami structure and use Deep-LASI to perform titrations, analyze protein conformational dynamics and demonstrate its versatility for analyzing both total internal reflection fluorescence microscopy and confocal smFRET data.

Single-molecule spectroscopy has revolutionized how we investigate the mechanism of processes on the nanometer scale. In particular, optical fluorescence imaging allows contact-free investigations of single, dynamic biomolecules, one at a time, in cells, membranes and in solutions. Single-molecule Förster Resonance Energy Transfer (smFRET) in combination with confocal microscopy or Total Internal Feflection Fluorescence (TIRF) microscopy probe distances on the nanometer scale (2.5–10 nm). While solution measurements can provide information on sub-millisecond dynamics, measurements with immobilized molecules give access to the temporal evolution of single molecules on the timescale of microseconds to minutes[1]. By removing ensemble averaging, it is possible to directly measure the underlying conformational states and molecular dynamics of biomolecules. Its ability to measure accurate distances and kinetics turned smFRET into a powerful tool for deciphering molecular interaction mechanisms and structures of biomolecules[1–3]. Typically, FRET experiments are performed using two colors and used to probe conformational distributions and distance changes. However, also other single-molecule approaches can be used to investigate small distance changes or interactions (e.g., Metal-Induced Energy Transfer (MIET)[4], Graphene Energy Transfer (GET)[5], or Protein-Induced Fluorescence Enhancement (PIFE)[6–8]).

When combining three- or more labels, multi-color FRET can probe molecular interactions between different binding partners and also measure multiple distances simultaneously, i.e. correlated motion within the same molecule[9–11]. However, multi-color analyses remain challenging. Quantitative smFRET data analysis is strongly hampered by experimental restrictions due to, for example, a low number of usable single molecule traces, data with a low signal-to-noise ratio (SNR), or short traces due to photochemistry. Overcoming these limitations requires large data volumes as very few molecules contain the desired information with suitable quality, which significantly increases

[1]Department of Chemistry and Center for NanoScience (CeNS) Ludwig-Maximilians-Universität München Butenandtstr. 5-13, 81377 Munich, Germany. ✉e-mail: evelyn.ploetz@lmu.de; d.lamb@lmu.de

the efforts involved in sorting through the data when performed manually. Low statistics result from various reasons including molecular events exhibiting slow kinetics or rare transition probability, insufficient labeling efficiency, low SNR, quick photobleaching or spurious background. In addition, arbitrary fluctuations due to unwanted interactions and/or aggregations between binding partners hamper a concise analysis of the underlying state and kinetics.

Various approaches have been developed to overcome these time-consuming burdens, employing user-defined thresholds on the channel count rate, signal-to-noise ratio, FRET values, FRET lifetime, and donor/acceptor correlation[12–19]. However, setting appropriate thresholds requires a substantial amount of expertise. Depending on the user, the data evaluation is prone to cognitive biases and poses a challenge to reproducible analysis results. Recently, software packages have been published that use deep-learning techniques to rapidly automate trace classification and keep user bias to a minimum[20–22]. In particular, Thomsen et al. comprehensively demonstrated that artificial neural networks could match manual classifications and even outperform conventional methods of commonly used programs to extract valid single-molecule FRET traces[22]. So far, deep learning has been solely applied to single-channel and two-color FRET data to categorize the time trajectories for downstream analysis. To study structural dynamics, reflected by changes in intensity and FRET efficiencies, the kinetics are then analyzed separately typically using Hidden Markov Models (HMMs)[23,24] approaches. Training an HMM requires knowledge of the number of states and modeling of the emission probabilities. Moreover, it assumes that the probability of a transition to the next state only depends on the current state. While the initial HMM settings are straightforward for simple systems, obtaining the optimal parameters for multi-color FRET becomes a challenging task. To date, only one software package, SMACKS[13], allows an ensemble HMM for three-color FRET data. As the complexity of the datasets grows, the effort and the required knowledge about the system also grow.

To alleviate the shortcomings of HMM analyses, the hybridization of HMMs with Deep Neural Networks (DNN) has gained popularity[25–29]. In contrast to HMMs, DNNs are capable of learning higher-order dependencies without prior assumptions about the number and properties of the states. A long-short-term memory (LSTM) neural network was developed to automate stoichiometry determination via photobleaching steps in fluorescence intensity traces[30]. However, the use of DNNs for extracting quantitative kinetic information from single-molecule data has not yet been explored.

Here, we present the Deep-Learning Assisted, Single-molecule Imaging (Deep-LASI) approach, an ensemble of DNNs with architectures specifically designed to perform a fully automated analysis of single-color traces as well as two-color and three-color single-molecule FRET data. Deep-LASI begins with raw intensity traces and provides corrected FRET efficiencies, state determination, and dwell times without any prior knowledge or assumptions about the system. It classifies each time trace into different categories, identifies which fluorophores are active in each frame, which is then used for determining FRET correction factors for spectral crosstalk, direct acceptor excitation and detection efficiency, and performs a state transition analysis of the different states in dynamic traces. Deep-LASI also includes optional number-of-state classifiers to estimate the actual number of observed states within one trace. Since the pre-trained neural networks operate locally on each trace, they do not neglect rare events, which would be missed in global analysis approaches. We benchmark the performance of Deep-LASI using ground truth simulations and experimental one-, two- and three-color data using an L-shaped DNA origami structure with tunable dynamic behavior[5,31]. The results are further compared to the manual evaluation of the data and the extracted dwell times obtained with HMM. Finally, we demonstrate the power of Deep-LASI with multiple applications: (1) titration

experiments, which would be unfeasible without Deep-LASI; (2) smFRET on a mitochondrial Hsp70 to extract substrate-specific dwell times and conformational states; and (3) the applicability of Deep-LASI to another experimental setup.

## Results
### The Deep-LASI approach

Deep-LASI utilizes an ensemble of pre-trained deep neural networks designed for the fully automated analysis of one-, two- and three-color single-molecule data including multi-color FRET correction and kinetic analyses (Fig. 1; Supplementary Note 1). The designed input for Deep-LASI is a single-molecule fluorescence intensity trace or traces measured directly using confocal microscopy or extracted from movies using wide-field or TIRF microscopy. In the case of two-color fluorescence data, continuous wave excitation or Alternating Laser EXcitation (ALEX) modalities can be analyzed. For three-color smFRET measurements, ALEX data is required. All available channels are fed into a combination of a Convolutional Neural Network (CNN) using the omniscale feature learning approach and a Long Short-Term Memory (LSTM) model (Supplementary Fig. 1.1).

Deep-LASI extracts spatial and temporal sequence features simultaneously and classifies every frame into a specific category (Fig. 1a). Building upon Deep-FRET for two-color FRET analysis[22], we separate the traces into six categories: dynamic, static, noisy, artifact, aggregate as well as photobleached (see Supplementary Note 2 for details). The total number of categories depends on the number of input channels, i.e. the number of dyes (and alternating light sources) used in the experiment. Traces containing random artifacts, aggregates, or high noise are excluded from further analyses. The final output of the state classifier provides an estimation of the probability for each category. The summed probabilities over all non-photobleached frames serve as confidence levels for each trace. Here, user-defined thresholds can be set to increase or decrease the tolerance towards non-ideal traces to be included in further analyses. In contrast to previous networks, Deep-LASI detects photobleaching events of individual dyes and, therefore, allows the calculation of correction factors obtainable for that molecule. Traces showing no apparent state transition are classified as static and can be included, e.g. in the final corrected FRET histograms.

All sections in each trajectory identified as dynamic are transferred to the state classifier network (Fig. 1b), which is designed to detect transitions based only on the intensity data and not via the FRET efficiency. The state classifier assigns every frame to one of the multiple states present in a dynamic trace section and again provides a confidence value of state occupancy that can be used for additional thresholding. Given the state transition classifications, a Transition Density Plot (TDP) is calculated and the kinetic rates of all identified states can be extracted by fitting the corresponding dwell-time distributions (Fig. 1c). Starting from trace extraction, the TDP marks the first necessary point of human intervention, i.e., the manual selection of state transitions and the fitting procedure. Thus, user bias is kept to a minimum. No assumptions are needed regarding the number of states, state-specific emission probabilities, or other settings required for conventional methods such as Hidden Markov Models (HMM). Of course, as for any deep-learning algorithm, the output of the analysis is dependent on the quality and appropriateness of the training data used. Depending on (1) the total number of frames, (2) the yield of valid frames, (3) the computer performance, and (4) the desired confidence threshold, a given dataset can be fully categorized on a time scale of 20–100 ms per trace.

### Training of Deep-LASI

To use Deep-LASI for analyzing single molecule data, we first trained the trace-classifier network on datasets appropriate for the corresponding network (i.e., one-color data, two-color data without ALEX,

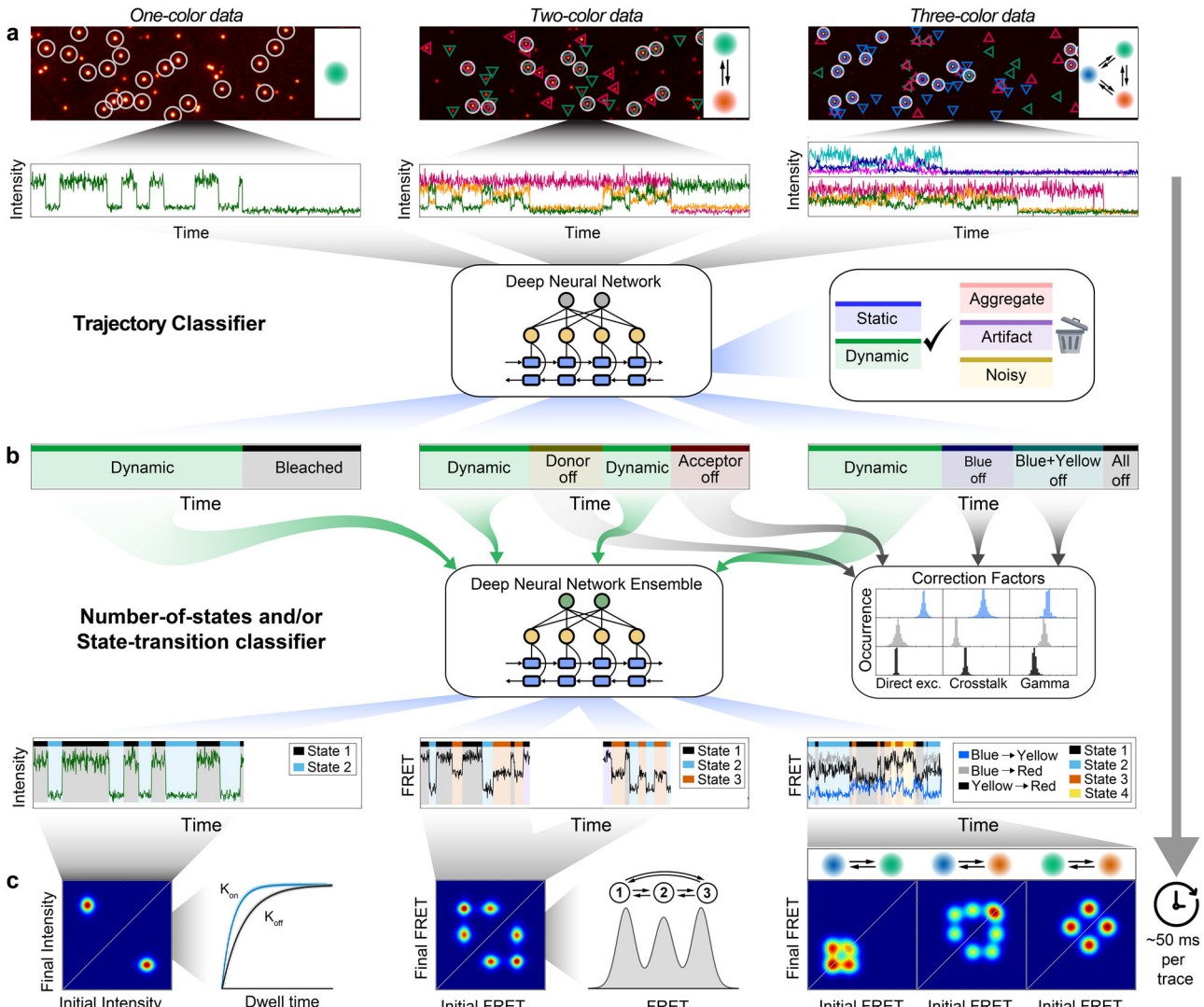

**Fig. 1 | Overview of data extraction, evaluation, and analysis using Deep-LASI.** **a** Single-molecule data of up to three separate channels after direct and alternating laser excitation are identified, extracted, and presorted for further analyses. Each frame within the time traces is classified into categories using a hybrid CNN-LSTM. **b** A second hybrid CNN-LSTM evaluates the kinetics and state information in the presorted data. The photobleaching information can be used for determining the correction factors to obtain accurate FRET values between two and three fluorophores. **c** Next, the interconversion rates between underlying states and absolute, distance-related FRET values are extracted from multi-color datasets.

two-color data with ALEX or three-color data with ALEX). As the noise sources in single-molecule fluorescence intensity data are well understood, simulated traces are well suited for training the neural network. In addition, it has the advantages of being able to minimize biases and quickly retrain neural network models to adjust for specific circumstances. The training datasets were designed to cover a wide range of experimental conditions and FRET efficiencies. Hence, no initial estimation of the number of states and expected FRET efficiencies are needed. A detailed description of the program architecture, simulations, training datasets and benchmarking can be found in the Methods section as well as in Supplementary Notes 1–4.

Deep-LASI contains a total of 16 pre-trained deep neural networks for state classification. Four models account for the classification and segmentation of time trajectories obtained from measurements using single-channel data acquisition, two-color FRET with continuous-wave excitation, two-color FRET with ALEX, and three-color FRET with ALEX. For each type of experiment, we provide three state-transition-classifiers trained on either two, three or four observed states, which take the output category dynamic as the input. Note that the acceptor intensity after direct excitation does not contain relevant kinetic

information and is not used in the state classifier networks. In addition, a deep neural network is provided that has been optimized for detecting the actual number of observed states and can be utilized for model selection. One network has been trained for each type of dataset (one-, two- and three-color data). The number-of-states neural networks are not essential in the automated analysis process but can serve as a safeguard against trajectories that may be out of the scope of the state transition classifiers.

## Performance of Deep-LASI
A common approach to benchmark classifier models is using ground truth labeled data and calculating confusion matrices, which summarize the correct and incorrect predictions. For every trained model (using ~ 200,000 traces), we generated approximately 20,000 new traces for testing, which were not part of the training dataset. Each of the validation datasets was then fed into the corresponding model. The output predictions were compared to the ground truth labels for every frame to obtain the percentage values of true positive, false positive and false negative classifications. All trace classifier models achieve a minimum combined precision of 97% in predicting smFRET

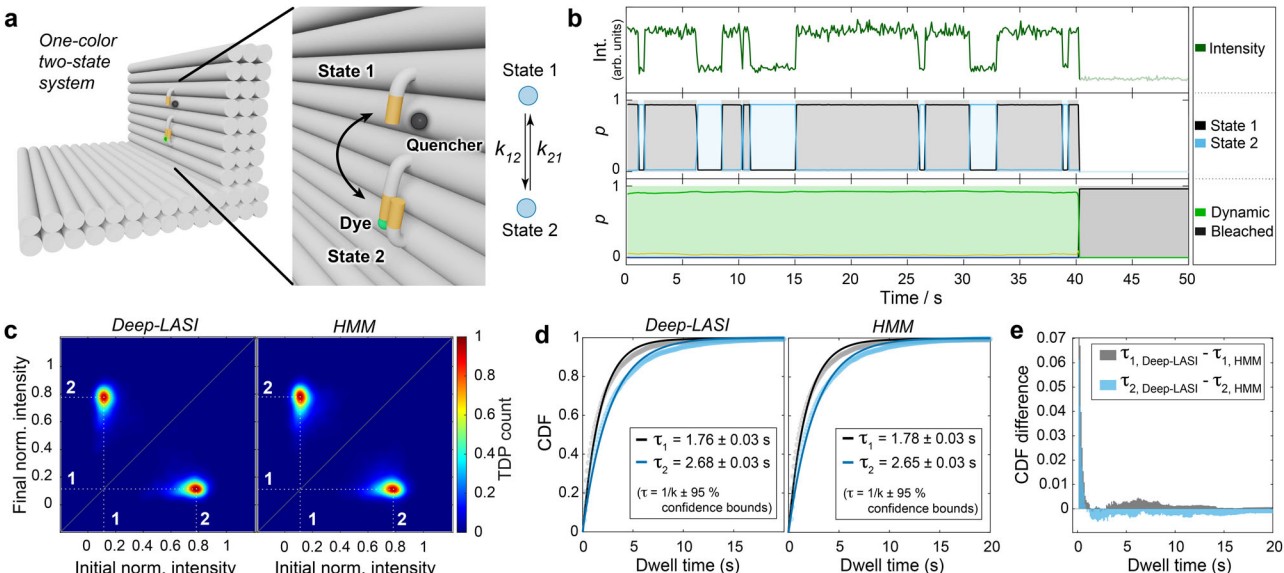

**Fig. 2 | State analysis of single-color single-molecule data. a** Sketch of the used L-shaped DNA origami structure with a single fluorophore (Cy3B) attached to a flexible tether, which changes position from state 1 → 2 at the rate $k_{12}$ and from state 2 → 1 at the rate $k_{21}$. The zoom-in shows the two single-stranded binding sites (orange) in close and distant proximity to a quencher dye (Atto647N) bound to the DNA origami structure. **b** Representative time transient for a DNA origami structure with 7.5 nt binding strands after classification and kinetic evaluation by Deep-LASI. **c** Transition-density plots depicting the interconversion events between the two detected states 1 and 2 after trace kinetics evaluation by Deep-LASI (left, number of transitions $n = 25{,}948$) and by Hidden-Markov Modeling (HMM) analysis (right,

number of transitions $n = 19{,}390$). Both approaches identify identical states. **d** Cumulative probability Distribution functions (CDFs) of the dwell times: The mono-exponential fits obtained by both methods reveal equivalent dwell times of approximately 1.75 and 2.65 s for the upper (State 1) and lower (State 2) binding sites, respectively. The errors in the dwell times are the 95% confidence intervals returned by the fitting procedure (estimated from the Jacobian matrix). **e** A comparison of the CDFs was determined using Deep-LASI and HMM. Deep-LASI is already sensitive at time scales on the order of the acquisition time. The average difference is less than 1% between both methods. Source data are provided as a Source Data file.

categories, i.e. static or dynamic, and 96% in predicting non-smFRET categories (Supplementary Figs. 3.1 and 3.2).

Our number-of-states and state-transition classifiers were benchmarked analogously. For the number-of-state classifiers, two states can be distinguished from multi-state trajectories with at least 98% precision whereas four states are predicted with the lowest precision of 86% for the single-channel model (Supplementary Fig. 3.3). For the state-transition classifiers, the states can be identified with accuracies of ≥ 98%, ≥ 90% and ≥ 78% for two-state, three-state and four-state models respectively (Supplementary Fig. 3.4). The comparison between all state-transition classifiers reveals a clear trend of decreasing accuracies with an increasing number of states and increasing accuracy with an increasing number of available channels. This is expected since a higher number of states have a larger probability of lower contrast, and a higher number of channels improves the robustness towards uncorrelated noise. Since confusion matrices do not reveal any underlying dependencies, we additionally benchmarked the state transition classifiers with HMM by calculating the precision of the state label prediction for a broad range of noise levels, FRET state differences and dynamic time scales (Supplementary Fig. 3.5). Overall, the performance of state classifiers is at least on par with HMM at low noise levels and outperforms HMM at high noise levels by up to 30%. To investigate the advantage of using the information in the entire dataset for the HMM analysis, we also compared the performance of Deep-LASI with a local and a global HMM on idealized synthetic data (Supplementary Fig. 3.6). Global HMM performs significantly better than local HMM in this case and is on par with Deep-LASI.

As a last test, we compared the performance of Deep-LASI with other kinetic analysis routines that have been recently published in a multi-laboratory study[32]. We chose to analyze the two-state datasets as these require no user input and the analysis can be performed without bias. Deep-LASI returned values corresponding to the ground truth for

the simulated dataset and close to the average values obtained for the experimental dataset (Supplementary Fig. 3.8).

## Deep-LASI analyses of DNA origami structures

Next, we benchmarked the potential of Deep-LASI to automatically analyze experimental data obtained from DNA origami structures. DNA origami is extensively used in bio-nanotechnology and has the advantage of being programmable with high precision and controllability. In particular, we choose an L-shaped DNA nanostructure with a dynamic, fluorescently labeled 19 nucleotide (nt) single-stranded DNA pointer. The geometry of the DNA structure was originally designed for measuring energy transfer to a graphene surface[5,31]. The single-stranded DNA pointer, along with two or three exchangeable docking strands of different complementary sequences, allows the number of states, position of the dyes, and kinetic rate to be programmed as desired. Hence, it is an ideal test system for measuring and extracting kinetic information from smFRET traces. FRET efficiencies and kinetic rates could be tuned by varying the position and complementary sequence length of binding strands on the DNA origami platform. We designed various DNA origami structures with one-, two-, and three-color labels and measured them on the single-molecule level.

In the first assay, we assessed Deep-LASI's capability to evaluate single-color data. For this, we probed one-color single-molecule kinetics where the flexible pointer was labeled with Cy3B at the 3'-end. Two complementary binding sites with 8 nt complementary nucleotides containing a 1 nt mismatch at the 5'-end (referred to as 7.5 nt) were placed about 6 nm below and above the pointer position (Fig. 2a). Binding occurred by spontaneous base-pairing to single-stranded protruding strands. A single red dye, Atto647N, acting as a quencher, was attached about 3 nm aside from the upper binding site (state 1). Figure 2b shows an exemplary intensity trajectory of Cy3B classified as dynamic until photobleaching was detected by the trace classifier with

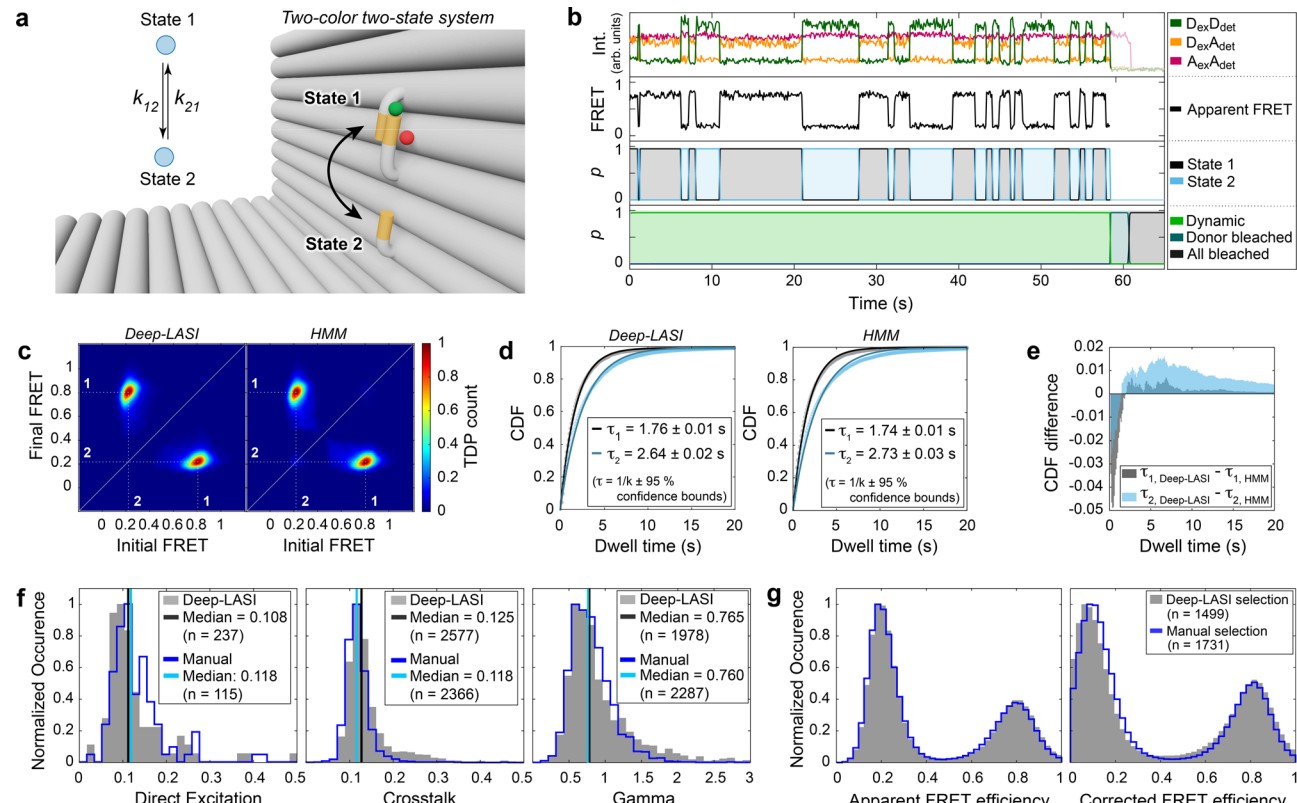

**Fig. 3 | Single-molecule analysis of two-color FRET data.** Experiments were performed with DNA origami structures exhibiting two FRET states. **a** Zoom-in of an L-shaped DNA origami structure labeled with Atto647N and Cy3B and corresponding kinetic scheme. The donor is attached to the flexible tether with a 7.5 nt overhang between the pointer and two single-stranded binding sites. FRET is expected between a high FRET state 1 (12 o'clock) and a low FRET state 2 (6 o'clock) interconverting at rates $k_{12}$ and $k_{21}$. **b** Representative single-molecule and apparent FRET trace after alternating red-yellow (RY) laser excitation. Deep-LASI classifies the trace and determines the underlying state for each frame. D: donor; A: Acceptor; ex: excitation; det: detection. **c** TDPs determined using Deep-LASI (left) and HMM (right) are shown revealing two interconverting states with apparent FRET values of 0.8 and 0.2. The two states are labeled in white. Total number of transitions: $n_{Deep-LASI} = 15,958$, $n_{HMM} = 21,243$. **d** CDFs extracted from the TDPs

shown in (**c**) and mono-exponential fits yield dwell times of 1.76 s and 2.64 s, respectively. The errors in the dwell times are the 95% confidence intervals returned by the fitting procedure (estimated from the Jacobian matrix). **e** A comparison of the cumulative dwell-time distribution determined using Deep-LASI - HMM for $\tau_1$ (gray) and $\tau_2$ (cyan). **f** Histograms of trace-wise determined correction factors for direct excitation, crosstalk and detection efficiency, either derived automatically by Deep-LASI (gray histograms, median in black) or determined manually (blue lines, median in cyan) (see Supplementary Note 5). **g** Apparent (left) and corrected (right) frame-wise smFRET efficiency histograms for 1499 dynamic traces from a total of 6100 traces. The states have corrected peak FRET efficiencies of 0.07 and 0.81. The histograms from traces selected by Deep-LASI are shown in gray and by manual selection in blue. Source data are provided as a Source Data file.

two corresponding states determined by the state classifier as the linker moves up and down.

We compared the results from Deep-LASI with a Hidden-Markov-Model analysis (HMM) trained on the same dataset. Since the state classifier does not directly predict a pre-trained intensity value for each state, the TDP was generated by averaging the normalized intensity between transitions. Both methods yield identical TDPs (Fig. 2c). The residence time of the DNA tether in both states was determined by fitting the cumulative dwell-time distribution functions (CDFs) derived from the state-classifier of Deep-LASI with a mono-exponential fit and compared to the results from HMM. The dwell times of 1.76 s versus 1.78 s (State 1) and 2.68 s versus 2.65 s (State 2) for Deep-LASI and HMM, respectively, are in excellent agreement (Fig. 2d). The differences between the CDFs obtained by Deep-LASI and HMM (Fig. 2e) indicate that Deep-LASI identifies fast transitions close to the frame time more frequently than HMM. The overall difference at longer dwell times remains well below 1%, which proves that Deep-LASI obtains identical results to HMM with negligible differences in the extracted rates. Interestingly, although the DNA binding strands are identical in sequence and length, there are clear differences in the dwell times. We attribute this to an inherent bias in the equilibrium position of the DNA pointer and non-symmetric, non-specific dye-origami interactions. In

addition, it is unlikely that the distance to each docking strand and potentially induced stress upon binding are identical for the two binding sites, even though the binding sequence is the same. We note that the kinetics we measure here are not directly comparable to other DNA-hybridization experiments due to both interacting DNA strands being tethered to the DNA origami platform. This leads to a high local concentration of the binder strand, and multiple dissociation and rebinding events can occur before the tether switches binding sites.

In the next step, we studied Deep-LASI's ability to deal with two-color data. We investigated two-color FRET assays with two states and compared the results with a pure manual evaluation of the same data. Here, both donor and acceptor signals from the same DNA origami sample system as shown in Fig. 2a were analyzed (Fig. 3a). TIRF measurements were performed using msALEX[33] yielding donor signal (Cy3B, Channel $D_{ex}D_{em}$), sensitized emission (Channel $D_{ex}A_{em}$) and acceptor signal (Atto647N, Channel $A_{ex}A_{em}$) to obtain information about acceptor photobleaching and direct excitation. Figure 3b shows a fully classified example trace with the signals on top and the derived FRET trace below. From the trace classifier, Deep-LASI identified dynamic sections and individual photobleaching events (Fig. 3b; bottom). The dynamic section was further classified in the state transition classifier according to their state occupancy using only the two

channels of the donor and acceptor intensity after donor excitation (Fig. 3b; middle). The channel of acceptor excitation and detection does not serve as input for the state transition classifier since it does not contain valuable kinetic information. From a total of 6100 recorded traces in the dataset, 1499 traces were classified as dynamic smFRET trajectories with at least one transition.

The same traces were also sorted manually and the 1731 selected dynamic traces were analyzed using HMM[34] (see Supplementary Note 5 for details). TDPs from the state transition classifier and from the HMM analysis are nearly identical (Fig. 3c). Also, the corresponding dwell times, determined via mono-exponential fits to the CDFs, are similar (Fig. 3d) and correspond to the expected dwell times of the one-color sample shown in Fig. 2 (~1.75 s for state 1 and 2.68 s for State 2). A comparison of the CDFs from Deep-LASI and HMM indicates that manually selected traces contained more fast transitions than the traces selected by Deep-LASI in this case (Fig. 3e). We looked into the differences between manually selected traces and traces selected by Deep-LASI. The most common classification discrepancies between the two are discussed in Supplementary Note 4.1. Based on individual example traces, we observed a stronger influence of the non-ideal behavior of the traces outside the regions of interest (e.g. a noisy ALEX signal or nonconstant signal intensities in photobleached regions of the trace) on Deep-LASI's classification compared to that of manual selection. We also compared the output of Deep-LASI with that of a global HMM analysis executed on the same dataset (Supplementary Fig. 4.2). As expected, the global HMM was prone to miss transitions due to slight heterogeneities in the dataset.

Next, we investigated how sensitive Deep-LASI is to the training dataset. Hence, we trained two additional classifier networks using newly simulated datasets. Details are given in Supplementary Note 4.2. The consistency between the differently trained neural networks is ~90%, similar to what would be expected from analysis run on the validation datasets (Supplementary Fig. 3.1c). Interestingly, the consistency between the different neural networks is higher than between two independent users (Supplementary Fig. 4.3b).

To determine the distance between both dyes in the two FRET states, the smFRET data needs to be corrected. Deep-LASI uses the frames classified as photobleached to automatically derive the correction factors necessary for an accurate FRET calculation[1,35,36]. In the manual analysis, the relevant regions are selected by hand (Fig. 3f, Supplementary Note 5). The correction factors agree within ~3%. Using the derived correction factors, the correct FRET efficiency is determined. The apparent (left) and corrected FRET histograms (right) of the Deep-LASI (gray histograms) and manually (blue lines) selected traces are shown in Fig. 3g. There is excellent agreement between the Deep-LASI and manually analyzed apparent FRET histograms. The difference between the corrected histograms is due to the difference in the correction factors determined and applied from the two analyses. In this case, as Deep-LASI classifies photobleaching on a per-frame basis, more frames can be used for determining the correction factors and are, thus, most likely, more accurate here. The corrected peak FRET efficiencies are 0.81 and 0.82 (State 1) and 0.08 and 0.14 (State 2) for Deep-LASI and manual evaluation, respectively, and correspond to distances of 53 and 53 Å, and 103 and 92 Å (assuming an $R_0$ of 68 Å[7]).

In the last step, we then tested the performance of Deep-LASI for analyzing three-color data by labeling the DNA origami structure with an additional blue dye, Atto488, at ~3 Å distance to the binding site for State 2 (Fig. 4a). The labeling sites of the yellow (Cy3b) and red (Atto647N) dyes were left unchanged to provide consistency with the previous two-color experiments. The use of three FRET pairs provides three distances simultaneously and allows the resolution of states that are degenerate for two-color FRET.

Using the six available intensity traces, each frame is categorized by the fluorophores that are active and whether the trace is static,

dynamic or should be discarded. As the acceptor intensity after acceptor excitation ($R_{ex}R_{em}$) does not contain valuable kinetic information, the other 5 intensity channels for dynamic traces (before photobleaching) are given as input for the state transition classifier (Fig. 4b). Movement of the flexible tether results in an anti-correlated change in the FRET efficiency of blue to yellow (BY) and yellow to red (YR), visible in the apparent FRET panel of the example trace in Fig. 4b. For each FRET pair, a TDP can be calculated, which allows the assignment of the state number to the actual FRET populations (Fig. 4c). Note, the apparent FRET efficiency of blue to red (BR) varies with the YR FRET efficiency due to the different energy transfer pathways taken upon blue excitation. Deep-LASI classifies a state regardless of which dye is undergoing a transition, i.e. the extracted dwell time distribution of a given state is the same for all FRET pairs when there is no overlap of multiple states in the TDP. The dwell times for states 1 and 2 match with those for the one-color and two-color samples, which indicates that the transition rates are not influenced by the acceptor dyes close by (Figs. 2d, 3d, Supplementary Fig. 6.1). From a total of 2545 recorded molecules, 581 were classified as valid, dynamic three-color FRET traces. The uncorrected, framewise smFRET histograms of BY, BR and YR FRET pairs are very similar to those from the 694 manually selected traces (Supplementary Fig. 4.4a). A detailed comparison between the manual analysis of the results from Deep-LASI is given in Supplementary Notes 4.3 and 4.4.

As for two-color FRET, Deep-LASI automatically determines all correction factors obtainable per trace depending on which dyes are photoactive. The results of the automated extraction of correction factors are summarized and compared to manually derived correction factors in Supplementary Fig. 4.4b. The corresponding apparent und state-wise, corrected FRET efficiency histograms for each FRET pair are shown in Fig. 4d. While the YR FRET efficiency can be directly calculated, the corrected BY and BR FRET efficiencies are subjected to higher uncertainties due to the large number of correction factors involved (see Supplementary Note 5). In particular, their dependency on the YR FRET efficiency leads to the broadening of the distributions. To minimize this influence, we perform the correction using the state-averaged FRET efficiencies. After correction, the FRET efficiencies of State 1 (0.81) and State 2 (0.08) for the YR FRET pair are virtually identical as for the two-color system. For the BY FRET pair, State 1 and State 2 correspond to peak FRET efficiencies of 0.36 and 0.81, respectively. As expected, the two populations of the apparent BR FRET efficiency merge into one static population in the corrected histogram with a peak FRET efficiency of 0.36.

To probe the performance of the kinetic analysis from Deep-LASI, we used the tunability of the L-shaped DNA origami structure to vary the timescale of the dynamics. In addition to the 7.5 nt binding sites (Fig. 4a–d), we measured three samples using binding sites of length 7 nt with a 1 nt mismatch (referred to as 6.5 nt), 7 nt, and 8 nt (Fig. 4e). The summary of all extracted dwell times (Fig. 4f, Supplementary Figure 6.1) shows an exponential increase in the dwell times of both states with increasing binding site lengths ranging from 0.33 s to 9.5 s. Considering the camera exposure times of 32 ms (6.5 nt), 50 ms (7 nt and 7.5 nt datasets) and 150 ms (8 nt dataset) and frame shift time of 2.2 ms, a dwell-time to frame-time ratio ranges from 9 (6.5 nt State 1) to 62 (8 nt, State 2).

To test Deep-LASI with more complex dynamics with multiple states, we constructed a three-state system with three-color labels using 7 nt binding strands at positions 6 and 12 o'clock and an additional 7.5 nt complementary binding strand at 9 o'clock (Fig. 5a). An example trace containing all possible transitions identified by Deep-LASI is shown in Fig. 5b. The TDP of the BY FRET pair (Fig. 5c, left panel) yields clearly distinguishable populations, while the TDP of the YR FRET pair (Fig. 5c, right panel) shows a degeneracy of state 3 transitions. Using the BY TDP, we determined the dwell time distributions with residence times between 0.65 s and 1.43 s (Supplementary

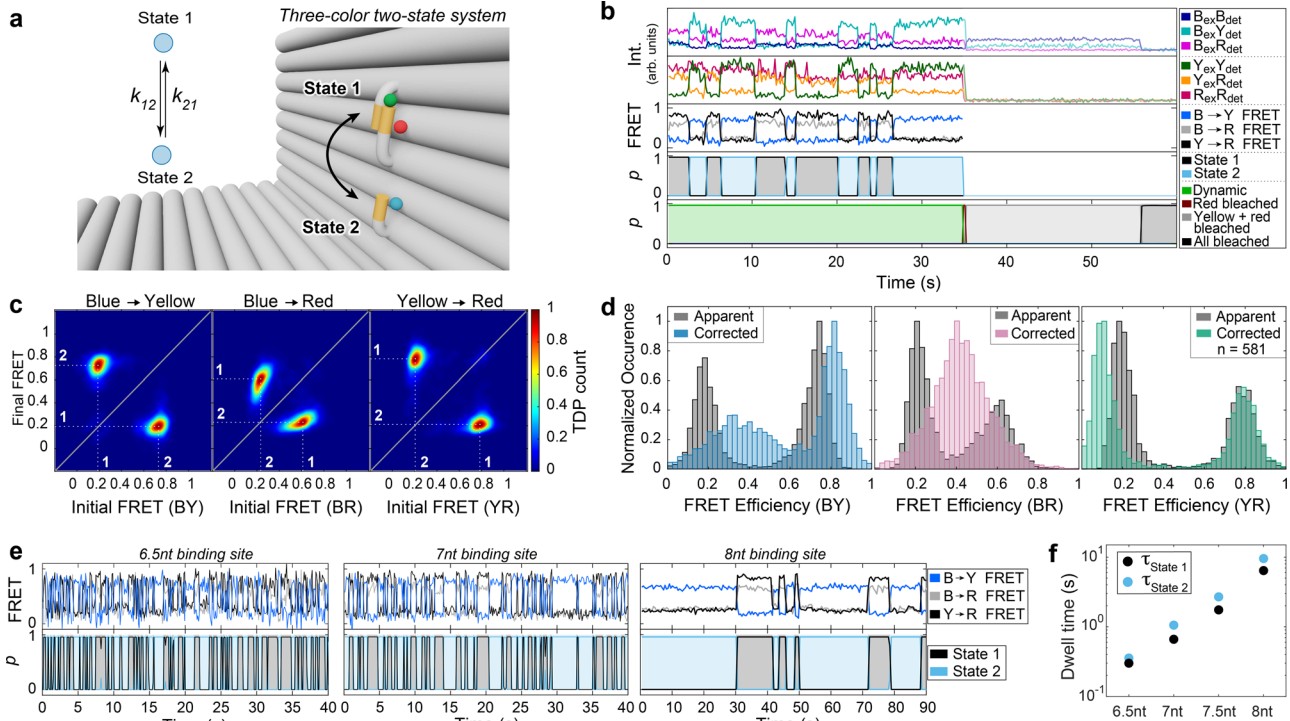

**Fig. 4 | Single-molecule analysis of three-color FRET data.** Experiments were performed on two-state DNA origami structures that were labeled with Atto647N, Cy3B and Atto488. While Cy3B is attached to a flexible tether, Atto647N and Atto488 are positioned close to the top (12 o'clock; state 1) and bottom (6 o'clock; state 2) binding sites, respectively. **a** Zoom-in of the L-shaped DNA origami structure and corresponding kinetic scheme. **b** Representative single-molecule intensity and FRET trajectories for binding sites with 7.5 nt overhang after alternating red-yellow-blue laser excitation. First panel: Intensities after blue excitation. Second panel: intensities after yellow and red excitation. Third panel: corresponding three-color FRET efficiencies. Fourth and fifth panels: Deep-LASI output for state transition and trace classification. B: blue; Y: yellow; R: red; ex: excitation; det: detection. **c** TDPs of the apparent FRET efficiency states reveal an apparent distance change for all three FRET pairs (BY (left), BR (middle), and YR channel (right)) with dwell times of 1.75 s and 2.69 s for the upper and lower binding site, respectively, nearly

identical to the two-color DNA origami structures (Fig. 3c). Total number of transitions: 5,013. **d** Frame-wise weighted state-wise apparent (gray) and corrected (color) smFRET efficiency histograms of the BY (left), BR (middle), and YR (right) FRET pairs. As expected, the accurate FRET efficiency of the BR pair is static (E = 0.36). As the position of Cy3B changes from state 1 to state 2, the accurate FRET efficiency changes from 0.36 to 0.81 (BY pair) and from 0.81 to 0.08 (YR pair). **e** Upper panel: Representative three-color smFRET traces for binding sites with 6.5 nt (7 nt with 1 nt mismatch), 7 nt and 8 nt overhangs after alternating RYB laser excitation. Bottom Panel: The corresponding state determined by Deep-LASI. **f** Extracted dwell times from mono-exponential fits for the lower (blue) and upper positions (black) for 6.5 nt ($\tau_1$: 0.31 s, $\tau_2$: 0.4 s), 7 nt ($\tau_1$: 0.66 s, $\tau_2$: 1.05 s), 7.5 nt ($\tau_1$: 1.75 s, $\tau_2$: 2.69 s) and 8 nt overhangs ($\tau_1$: 6.41 s, $\tau_2$: 9.54 s) (see Supplementary Figure 6.1 for more details). nt: nucleotides. Source data are provided as a Source Data file.

Figure 6.2). The three states are well-resolved in the framewise apparent BY FRET histogram, while state 2 and state 3 are degenerate for the BR and YR FRET pairs (Fig. 5c). Applying all correction factors yields peak YR FRET efficiencies of 0.81 (state 1), 0.08 (state 2) and 0.19 (state 3). Upon correction, States 1 and 3 in the BY FRET histogram merge into a broad degenerate FRET population. However, using the state information for all three fluorophores allows us to separate out the BY FRET histograms of the individual states.

For three-color FRET, the corrected BY and BR FRET efficiencies depend on the YR FRET efficiency, and the additional corrections broaden the population. However, even though the data may be noisier, three-color experiments contain additional information, which typically allows one to resolve states that are degenerate in two-color experiments. This is exemplified in two-color FRET experiments on the same construct missing the blue fluorophore near the 6 o'clock binding site (Supplementary Note 6.3). For distinguishable states, the determined corrected FRET efficiencies and kinetic rates from two- and three-color experiments are the same. However, three-color FRET experiments enable the lifting of this degeneracy between states 2 and 3. To minimize the influence of the increased noise in three-color experiments, it is advantageous to analyze the data in proximity ratio and only convert it to corrected FRET efficiencies when necessary[10]. Deep-LASI can rapidly classify a large number of molecules and quickly

provide an overview of multi-state dynamics with easy access to the kinetic information.

## Further applications of Deep-LASI

After extensive benchmarking, we applied Deep-LASI to various single-molecule datasets originating from biophysical assays, protein samples and experimental systems beyond TIRF microscopy. With the speedup in analysis time from days to minutes, experiments become possible that would have been unthinkable when performing the analysis manually. One example is a titration experiment where the biochemical conditions are varied. Here, we measured the influence of glycerol concentration on the dynamics of the 3-colored L-shaped DNA origami introduced in Fig. 4a with 7.5 nt overhangs. Interestingly, we observed a decrease in residence time in both states with increasing glycerol concentrations (Fig. 6a, b). Dwell times start at 1.75 s (state 1) and 2.69 s (state 2) for pure imaging buffer and decrease down to 0.62 s and 0.85 s in buffer containing 30% (v/v) glycerol. The multi-fold increase in binding kinetics can be explained by a reported destabilization of base-pairing due to changes in the ssDNA hydration shell[37] and concomitantly disturbed hydrogen bonding due to the osmolyte-DNA interaction. The melting enthalpy and melting temperature decreases linearly with glycerol concentration at about 0.2 °C per % (v/v)[38,39] in line with our observations (Fig. 5b). With Deep-LASI at hand,

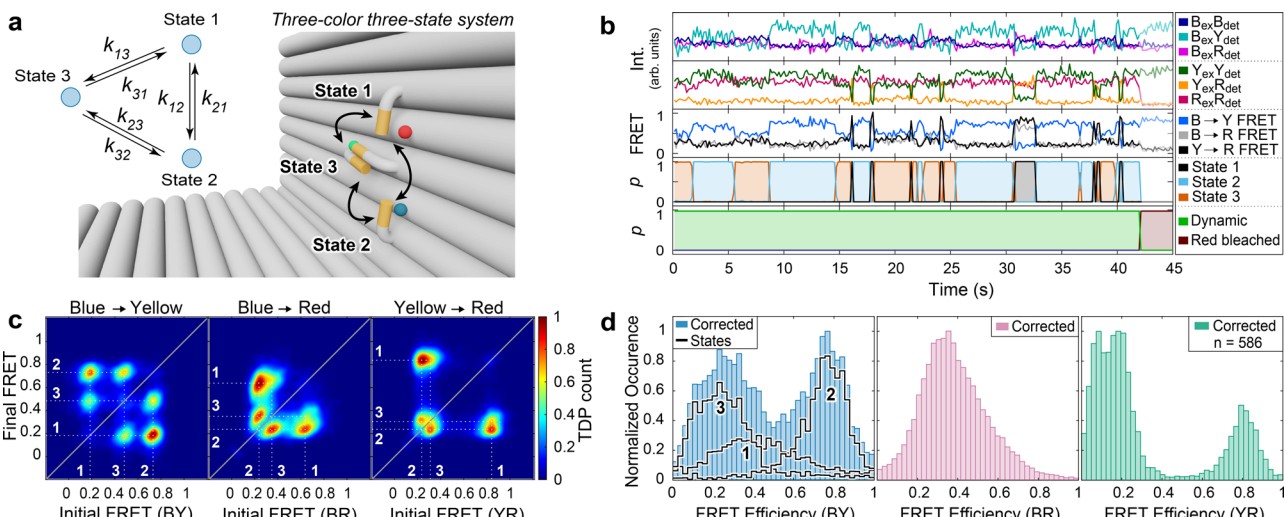

**Fig. 5 | Single-molecule analysis of three-state, three-color FRET data. a** Zoom-in of the L-shaped DNA origami structure with an additional binding site for the tether (state 3 at 9 o'clock) and corresponding kinetic scheme. **b** A representative single-molecule three-color FRET trace and apparent FRET for the 3-state system. The upper panel shows the intensity in the blue, yellow and red channels after blue excitation. The second panel shows the intensity in the yellow and red channels after yellow excitation and the red intensity after red excitation. The middle panel shows the corresponding FRET efficiencies for the three dye pairs. The fourth and fifth panels show the output of the Deep-LASI analysis for state transition and trace classification, respectively. B: blue; Y: yellow; R: red; ex: excitation; det: detection.

**c** Transition density plots of the apparent FRET efficiency states are shown for each FRET pair revealing an interconversion between 3 binding sites. Total number of transitions: $n = 17,136$. **d** Frame-wise weighted, state-wise corrected smFRET efficiency histograms. Corrected, distance-related FRET values are best resolved for the YR pair, showing three populations at 0.81, 0.19 and 0.09. The BY FRET shows one population at 0.8, corresponding to state 2, and a broad population at 0.3 for states 1 and 3. Individually-corrected states are indicated with the highlighted lines, showing the actual BY FRET efficiencies of state 1 (0.4) and state 3 (0.21). The apparent FRET states for the BR channel merge into one broad, static state with a value of 0.35. Source data are provided as a Source Data file.

local screening and time-consuming parameterization of imaging conditions become feasible.

Next, we applied Deep-LASI to smFRET measurements on proteins. We previously used dual-color FRET studies to probe the nucleotide-dependent conformational states[40] of Ssc1, a mitochondrial heat-shock protein Hsp70 in yeast. By fluorescently labeling both the nucleotide-binding domain and the substrate-binding domain, we investigated the influence of ADP on the inter-domain separation via smFRET. As the proteins were immobilized by incorporation in vesicles, a photostabilization buffer could not be used for the experiments. Hence, the signal-to-noise ratio in these experiments is lower than those exhibited by the photostabilized DNA origami structures. A comparison of traces selected manually and/or by the Deep-LASI analysis is discussed in Supplementary Note 6.5. For the different ADP concentrations, Deep-LASI identifies the underlying FRET states in line with the manually evaluated data[40] (Fig. 6c). It correctly identifies transitions between two distinct states, a loosely docked conformation with high FRET efficiency (E = 81%) and a separated undocked state (E = 50%), as shown in Fig. 6d. The automated data analysis of Deep-LASI confirmed the ADP-dependent kinetics of the domain sensor in good agreement with previous, manually evaluated results[40] (Fig. 6e, f). This demonstrates the proficiency of Deep-LASI for unsupervised data evaluation of smFRET data on proteins.

Finally, we tested the automated analysis of Deep-LASI applied to a different microscopy approach for smFRET, i.e. confocal single-molecule data on immobilized molecules that can be collected with microsecond time resolution. We chose the same double-labeled DNA origami structure introduced in Fig. 2a but with different combinations of docking strands. For these constructs, the measured FRET efficiencies will be the same but with different dynamics. By changing the hybridization length or by adding mismatched bases in the docking strand, the interaction time of each binding site can be tuned individually from ~1 ms to 10 s of seconds by adjusting the stabilization energy of DNA hybridization. As expected, shorter hybridization sequences lead to fast dynamics. Figure 6g shows a representative

intensity trajectory of a DNA origami structure (containing 6 nt complementary overhangs) that was classified as dynamic and the corresponding predictions of the state classifier. Although the unquenched state (state 2) shows a high variance in intensity, the state classifier predicts transitions with high accuracy and confidence. In the case of the 5 nt complementary overhangs, the dwell times approach 1 ms (Fig. 6h), and the output probability, $p$, of the state classifier decreases significantly due to the low signal-to-noise ratio of the trace. Thus, the probability value is an important parameter indicating the confidence the state classifier has in the assignment of the state and can be used as a threshold. Figure 6i (colored symbols) compares the mean dwell times extracted by Deep-LASI for all the confocal datasets with the results obtained by a newly developed shrinking-gate fluorescence correlation spectroscopy (sg-FCS) approach[41]. In sg-FCS, a pulsed light source is used such that the fluorescence lifetime information can be incorporated into the analysis. By shrinking the analysis window of photons based on their detected arrival time after excitation, we vary the relative brightness of two species with different fluorescence lifetimes (e.g. the low FRET and high FRET states). This leads to a robust extraction of the kinetic rates between the two states from the autocorrelation analysis of the FCS data. For all binding site combinations with 6 nt to 7 nt complementary overhangs, dwell times obtained by both methods are in excellent agreement. The largest deviation was found for the 6 nt binding sites in the asymmetric 6 nt/7 nt sample (Fig. 6i, purple) (a factor of 2) where there is large heterogeneity and limited statistics[41]. The dwell times for the sample with 5 nt complementary overhangs follow the exponential trend observed for longer binding sites but the binning of 0.6 ms, together with the resulting low signal-to-noise ratio, reach the current limit of Deep-LASI's state classifier. For completeness, we have included the results from Fig. 4e, f in Fig. 6i (gray triangles). There is a shift in dwell times between TIRF and confocal data due to the different temperatures of the two laboratories ( ~19 °C confocal, ~22 °C TIRF, see Supplementary Note 6.4). Lower temperatures lead to a higher standard free energy and concomitantly longer binding time[42,43]. In the case of the 6.5 nt

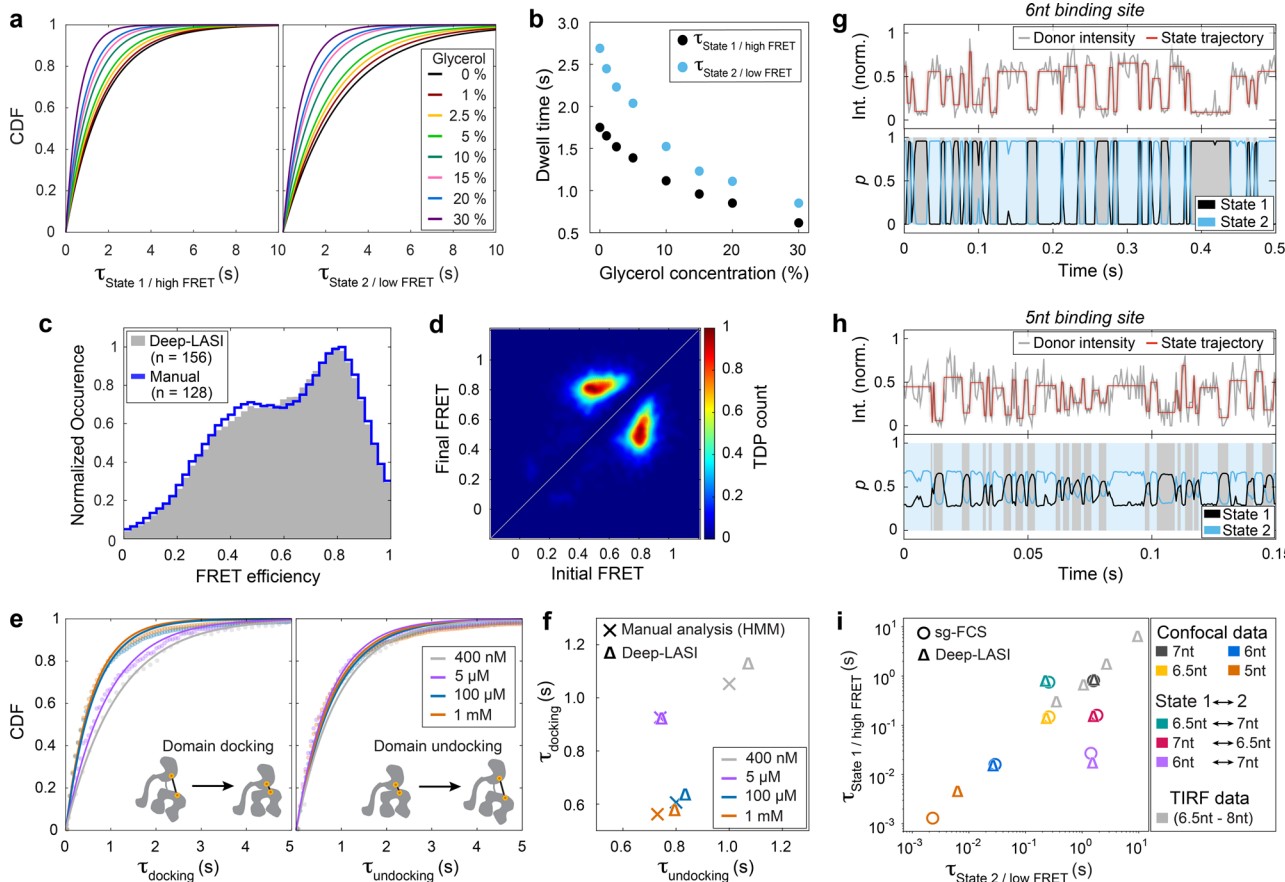

**Fig. 6 | Use of Deep-LASI on titration experiments, protein data, and confocal data. a, b** 3cFRET: Tuning the dissociation thermodynamics between protruding ssDNA strands by osmolytes. **a** CDFs of the dwell times, assessed through mono-exponential fits, for state 1 (left) and state 2 (right) of the L-shaped DNA origami structure from Fig. 4a decrease with increasing glycerol concentration. **b** Dependence of dwell times for both states versus glycerol concentration. **c–f** 2cFRET: Probing domain-domain interactions in Ssc1, a mitochondrial Hsp70. **c** Frame-wise smFRET distributions of Hsp70 molecules in the presence of 1 mM ADP classified as dynamic by Deep-LASI (gray) and evaluated manually (blue) from a total of 3534 traces. **d** The TDP generated by Deep-LASI aligns with the data plotted in (**c**), illustrating the interconversion between the undocked (~0.5) and docked (~0.8) conformations. Total number of transitions, *n* = 3914. **e** CDFs of the dwell times and mono-exponential fits to the dwell time distributions derived by Deep-LASI for domain docking (left panel) and domain undocking (right panel) depending on the ADP concentration. **f** Comparison between average dwell times extracted by Deep-

LASI (triangles) and by manual evaluation (crosses) using HMM. Deep-LASI matches the published trend with similar dwell times[40]. **g–i** 1c-FRET: Deep-LASI analysis of ssDNA binding kinetics observed via confocal microscopy. **g** Confocal trace (with 2 ms binning) of the DNA origami structure from Fig. 2a with 6 nt binding sites and corresponding states predicted with high confidence. **h** Confocal trace (with 0.6 ms binning) of a DNA origami structure with 5 nt binding sites and predicted states. Due to the low SNR of the data, the confidence output of Deep-LASI reaches its lower limit. **i** Mean dwell times obtained from confocal data for various binding site lengths analyzed by sg-FCS[41] (circles) and Deep-LASI (triangles). The results align well, except for dwell times extracted from the 5 nt sample, which was predicted with a low confidence distribution due to low SNR and a limited amount of information in the one-channel input. Dwell times obtained from TIRF data are displayed in light gray for comparison. nt: nucleotides. Source data are provided as a Source Data file.

binding sites sample (Fig. 6i, yellow), lower dwell times are consistently observed for the TIRF data. This discrepancy is due to the difference in temporal resolution of the two measurements (2 ms for confocal vs 30 ms for TIRF). The lower temporal resolution of the TIRF measurements led to a higher probability of fast transitions being averaged out and an underestimation of the actual transition time. This is a limitation of the real experimental data and is not attributable to Deep-LASI. On the contrary, Deep-LASI can back-trace shortcomings of either technique, identify rare events and monitor conformational changes over several time scales in an unsupervised manner.

## Discussion

Deep-LASI is a deep-learning algorithm for the rapid and fully automated analysis of one-, two- and three-color single-molecule assays. Employing state-of-the-art neural network architectures optimized for time series data, we extended the classification of two-color FRET trajectories to include single- and three-color data, analyzed the

photobleaching information and incorporated a full state transition classification.

The utilization of deep-learning approaches for single-molecule analysis comes with both advantages and potential pitfalls. One major advantage is the ability of neural networks to capture intricate temporal dependencies and complex patterns in time-series data. This allows for improved classification accuracy and the identification of subtle transitions or states that may be challenging to discern using traditional analysis methods. Additionally, deep-learning models can learn from large amounts of data, reducing the dependence on prior assumptions that may introduce user bias.

It is essential to consider potential pitfalls when using deep neural networks for single molecule analysis. One challenge is the interpretability of the neural networks' decisions. While mathematical models and simpler thresholding techniques introduce user bias, they provide explicit confidence levels or probabilities derived from the user's modeling choices. In contrast, the output generated by neural

networks can be viewed as an artificial confidence level, minimizing user bias and increasing the consistency of the analysis. However, potential unknown biases inherent in the network itself may be introduced. Although neural networks can demonstrate high accuracy on validation datasets, understanding the underlying features and mechanisms influencing their predictions can be more challenging compared to conventional methods with explicit assumptions.

Ideal single-molecule traces are straightforward to model, and Deep-LASI is trained to be stringent when encountering non-ideal traces at a high confidence threshold. Hence, the default output of Deep-LASI when analyzing data with poor SNR is to discard the majority of the traces. By adjusting this threshold, users can instantaneously modify the accepted traces and monitor changes in the final results. This approach ensures a very low false positive classification rate when using a high confidence threshold and allows for gradual threshold reduction to increase statistical coverage. However, the outcome should be continuously monitored by the user. This iterative process effectively balances stringent classification and the need for increased statistical robustness in the analysis of new datasets.

Furthermore, Deep-LASI offers an advantage in terms of interpretability as it is trained solely on editable and extendable simulations. This characteristic provides users with greater control and knowledge over potential biases and enables them to tailor Deep-LASI to a wide range of experimental conditions. It is important to note that neural networks are data-driven models and heavily rely on the quality and representativeness of the training data. Therefore, careful consideration must be given to curating the training dataset to avoid biases and ensure the generalizability of the model to diverse experimental conditions. Regular validation and testing using independent datasets are crucial steps to assess the robustness and reliability of the model's performance. In addition, when measuring an unknown experimental system for the first time, it is helpful to visually inspect the traces that are being discarded to verify that the classification is still reasonable. By following these practices, researchers can enhance the trustworthiness and applicability of Deep-LASI in real-world scenarios.

In conclusion, Deep-LASI addresses the need for rapid, high-throughput screening of fluorescence intensity trajectories. This opens new possibilities for single-molecule assays and enables a timely analysis of complex experimental approaches thanks to the efficient and retrainable neural network architecture of Deep-LASI. It has a high potential for applications in a myriad of fields including biotheranostics, sensing, DNA barcoding, proteomics and single-molecule protein sequencing. We envision that deep-learning approaches, along with single-molecule sensitivity, will dramatically assist and accelerate analytics and be indispensable in the future.

## Methods
### Chemicals
Chemicals were purchased from Sigma-Aldrich and used without further purification, if not stated otherwise. Chemicals include acetic acid, agarose, ammonium persulfate, (3-aminopropyl-) triethoxysilane (APTES), biotin-poly(ethylene glycol)-silane (biotin-PEG, MW3000, PG2-BNSL-3k, Nanocs, NY; USA), bovine serum albumin (BSA; New England Biolabs, Ipswich, MA, USA), Blue Juice gel loading buffer (ThermoFisher Scientific), ethylene-diamine-tetraacetic acid sodium salt dehydrate (EDTA-Na$_2$ × 2H$_2$O), glycerol, magnesium chloride (MgCl$_2$ × 6H$_2$O), 2-[methoxy(polyethyleneeoxy)propyl]trimethoxy-silane (mPEG, #AB111226, abcr; Germany), phosphate-buffered saline (PBS), protocatechuate 3,4-dioxygenase from Pseudomonas sp. (PCD), protocatechuic acid (PCA), streptavidin, sodium chloride, Tris base, Tris HCl, and 6-hydroxy-2,5,7,8-tetramethylchroman-2-carboxylic acid (Trolox) and beta-mercaptoethanol (βME).

All unmodified staple strands (Supplementary Note 7, Supplementary Table 7.2) used for DNA origami structure folding are commercially available and were purchased from Integrated DNA Technologies®. Staple strands with modifications (Supplementary Tables 7.3 and 7.4) were obtained from Biomers (Supplementary Table 7.3: Biotin; Supplementary Table 7.4: Atto488) and Eurofines Genomics (Supplementary Table 7.4: binding sites, Cy3b and Atto647N).

### DNA origami structures: assembly, purification and characterization
Preparation of the L-shaped DNA origami structures follows the procedures described previously by Tinnefeld et al.[5,31]. In brief, the L-shaped DNA origami structures were folded with a 10-fold excess of 252 different, unmodified and labeled oligonucleotides to the complimentary 8064 bp scaffold strand in folding buffer, which contained 40 mM Tris base, 20 mM acetic acid, 20 mM MgCl$_2$ × 6 H$_2$O, and 1 mM EDTA-Na$_2$ × 2 H$_2$O. A complete list with sequences of oligonucleotides used for producing the DNA origami structure is given in Supplementary Note 7. For folding, a nonlinear thermal annealing ramp over 16 hours was used[44].

After folding, the DNA origami solution was cleaned via gel electrophoresis in 50 mL 1.5% agarose-gel containing 1× gel buffer (40 mM Tris base, 20 mM acetic acid, 12 mM MgCl$_2$ × 6 H$_2$O, and 1 mM EDTA-Na$_2$ × 2 H$_2$O). The gel pockets were filled with a solution of 1× Blue Juice gel loading buffer and the DNA origami solution. The ice-cooled gel was run for 2 h at 60 V. When samples were to be recovered from the gel, the staining step was omitted and the Cy3b fluorescence was used instead to identify the correct DNA origami structures. Gel extraction was performed via cutting with a scalpel and squeezing the gel with a Parafilm® (Bernis®) wrapped glass slide. The concentration was determined by absorption spectroscopy on a NanoDrop 2000 device (ThermoFisher Scientific). Purified DNA origami structures were kept in storage buffer, i.e. in 1× TAE buffer (40 mM Tris base, 20 mM acetic acid and 1 mM EDTA-Na$_2$ × 2H$_2$O) with 12.5 mM MgCl$_2$ × 6 H$_2$O (pH = 8.4).

The correct folding of the DNA origami structures was confirmed using atomic force microscopy and transition electron microscopy (see Supplementary Figure 7.2).

### Sample preparation for multicolor prism-type TIRF experiments
Labeled DNA origami molecules were immobilized in flow channels formed between a coverslip and a surface-functionalized quartz prism. The surfaces were sandwiched on top of each other and sealed by a molten, pre-cut Nesco film (Nesco) channel. The employed prism surface was functionalized before with a biotin-PEG/mPEG coating to achieve surface passivation and prevent unspecific binding. Before the TIRF experiments, the prisms were first flushed with PBS and then incubated with a streptavidin solution (0.2 mg/mL) for 15 min. Afterwards, the sample holder was washed 3× with PBS to remove free streptavidin and then with storage buffer (1× TAE, 12.5 mM MgCl$_2$, pH = 8.4). Next, the DNA origami sample was diluted to 40 pM in storage buffer, added to the flow chamber and immobilized to the prism surface via the biotin-streptavidin linkage. After 5 min, untethered DNA origami structures were removed by rinsing the chamber 3× with storage buffer. Next, the attached fluorophores on the DNA origami structure were photostabilized by a combination of ROXS and an oxygen scavenging system based on PCA/PCD[45]. The photostabilization buffer was mixed as follows: 1 μL of 100 mM Trolox/Ethanol solution was added to 97 μL storage buffer. The sample was then aged using a UV Lamp (M&S Laborgeräte GmbH, UVAC-6U, 2 × 6 W; 254 and 366 nm) until an equal ratio of Trolox and Trolox-quinone was formed (typically 6 minutes)[46]. Immediately before starting the TIRF experiments, 1 μL of 100 mM PCA in methanol and 1 μL of 100 mM PCD solution (50% glycerol, 50 mM KCl, 100 mM Tris HCl, 1 mM EDTA-Na$_2$ × 2H$_2$O, pH8) was added to the total volume. In the case of samples containing Atto488, 1 μL of 14.3 M βME was added to the

photostabilization buffer. The sample chamber was flushed with photostabilization buffer and sealed to allow for enzymatic oxygen depletion. A minimum of 5 minutes waiting time preceded the experimental recordings. The photostabilization buffer was refreshed every 45 minutes until the end of the experiment.

All two- and three-color FRET experiments were carried out using msALEX[33], i.e. two- or three excitation lasers were alternated frame-wise. The lasers of different excitation wavelengths were synchronized using an acousto-optical filter (OPTO-ELECTRONIC, France) with the camera frame rate using an FPGA that synchronizes the excitation and simultaneous detection on the EMCCD cameras at 32 ms, 50 ms or 150 ms exposure times (depending on the sample) for 2000 (two-color) and 2400 (three-color) frames. The laser powers were set to 28 mW (0.022 mm², 491 nm), 16 mW (0.040 mm², 561 nm) and 10 mW (0.022 mm², 640 nm) for B-Y-R excitation.

### Multi-color TIRF setup

Single-pair FRET experiments on surface-immobilized DNA origami structures were carried out on a home-built TIRF microscope with prism-type excitation as previously published[47]. Three laser sources (Cobolt, Solna; Sweden) at 491 nm, 561 nm and 640 nm are available, and used for triple-color TIRF experiments with an alternation rate of 27 Hz (including a 2.2 ms frame transfer rate) between the B-Y-R laser excitation. The resulting emission was collected by a 60×water immersion objective (60×/1.27 WI Plan Apo IR, Nikon), cleaned up with a notch filter (Stopline® Notch 488/647, AHF), and the red emission was separated from the blue/yellow emission by a dichroic mirror (630DCXR AHF; Germany) followed by separation of the blue and yellow emission (560DCXR AHF). The emission was spectrally filtered (AHF Analysentechnik, Tübingen, Germany) for the blue (ET525/50), yellow (HQ595/50) and red (ET685/40) collection channels and afterwards detected on three EMCCD cameras (Andor iXon (1×)/iXon Ultra (2×), Andor Technologies, Belfast; UK) via the supplier's software Andor Solis (Version 4.29.30005.0; Oxford Instruments). Synchronization and alternation of the exciting laser sources, as well as the frame-wise data acquisition on three separate cameras, were achieved using a LabView-written program that controls a field programmable gate array (FPGA; NI cRIO-9073). While the program starts the measurement, the FPGA synchronizes the execution of the hardware via TTL pulses, i.e. it controls switching on/off the excitation sources by direct modulation of the AOTF (491, 561, 640 nm), while simultaneously starting the data acquisition by the three cameras. The videos were analyzed afterward by a custom-written MATLAB program (Version 9.13.0.2166757; Mathworks, Massachusetts, USA).

### Single-molecule data analysis, data evaluation and representation

Time traces of individual, fluorescently labeled DNA origami structures were extracted from measurements using one, two or three cameras for one-, two- and three-color experiments, respectively, using Deep-LASI. Deep-LASI is written in MATLAB (Version 9.13.0.2166757; Mathworks, Massachusetts, USA) and uses neural networks trained with the Python library TensorFlow (Version 2.8.0). All raw data were recorded by EMCCD cameras (iXon 897, i.e. frames with 512 × 512 pixels containing fluorescence intensity information) and stored as TIFF stacks using the supplier's software Andor Solis. The resulting traces are then analyzed either using the pre-trained neural networks (Supplementary Notes 2, 3) or manually (Supplementary Note 5). The regions of single-molecule traces that were classified as dynamic with photoactive fluorophores were selected for downstream analysis. In the automated analysis procedure, the state transitions and state dwell times were predicted by a neural network model. All manually selected traces were analyzed using Hidden Markov Models, locally fit to each intensity trace (1-color data) or FRET trace (2-color data) assuming two states with Gaussian emission distribution functions and using the

Baum-Welch algorithm. The Gaussian emission distribution functions serve as the prior for the HMM, which are iteratively updated during the analysis. The convergence threshold was set to $10^{-9}$ and the maximum number of iterations was set to $10^8$. All predicted transitions were extracted from a transition density plot and the corresponding dwell times were fit to an exponential function. All correction factors for calculating the corrected FRET efficiency were determined using the manual or automated classification of photobleaching steps. All employed methods, automated and manual, were performed using the Deep-LASI user interface. Final panels were all presented using MATLAB 2022b (Version 9.13.0.2166757; Mathworks, Massachusetts, USA), exported as vector graphics, and assembled into figures using Adobe Illustrator CS2022 (Adobe Inc.; USA). 3D representations of the DNA origami structures were rendered in Blender (Version 2.93.6), and further assembled and labeled in Illustrator. The AFM images were plotted using JPK Desktop Software (Version 6.1.200 A).

### Statistics & reproducibility

For training the neural network, we used a simulated dataset with ~200,000 traces as it is sufficient to cover an extensive range of realistic experimental parameters and thereby avoid any bias in the analysis. This includes FRET efficiencies between 0.01 and 0.99, dwell times of 1 to 100 frames and SNR of ~0.3 to 50. Experimentally, we typically measured 100 movies for each condition, as this usually generates several thousand acceptable traces. The full datasets were analyzed. The program, as part of its function, determines which intensity traces are suitable for further analysis. The computer selection was tested against simulated traces as well as compared with human analyses.

The Deep-LASI software was trained on three independently generated datasets. Deep-LASI was also compared with two users who manually analyzed the same datasets. The number of states, FRET efficiency histograms and kinetic rates extracted from the different analyses are consistent and, when available, are within the confidence intervals from the fits. Experiments were not randomized. The researchers were not blinded as knowledge regarding the sample did not influence the manual selection or analysis of the data. For the neural network, the advantage is that it operates only based on the data that it has been trained with. Hence, blinding is not applicable.

For analysis of the dwell-time distributions, a mono-exponential function was fit to the cumulative distribution function in MATLAB 2022b. The optimal fit values, along with the 95% confidence intervals, are given in the text and figures.

### Reporting summary

Further information on research design is available in the Nature Portfolio Reporting Summary linked to this article.

## Data availability

The data for all figures and all supplementary figures have been deposited in the Zenodo database[48] [https://zenodo.org/record/7561162], with the exception of previously published data (HSP70 SSC1 and DNA origami confocal data in Fig. 6). Source data are provided with this paper.

## Code availability

The program is available on GitLab [https://gitlab.com/simon71/deeplasi]. Extensive documentation for the Deep-LASI software package can be found at https://deep-lasi-tutorial.readthedocs.io/en/latest/index.html.

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

## Acknowledgements

We thank Julian Heeg for helping with data collection. We thankfully acknowledge the financial support of the Deutsche Forschungsgemeinschaft (DFG, German Research Foundation) – Project-ID 201269156 – SFB 1032 Project B03 (to D.C.L.), Project A13 (to P.T.), individual grants to PL696/4-1 (to E.P.), TI 329/9-2, project number 267681426, TI 329/14-1, and TI 329/15-1 (to P.T.) and Germany's Excellence Strategy – EXC 089/1 – 390776260. D.C.L. and P.T. gratefully acknowledge funding from the Federal Ministry of Education and Research (BMBF) and the Free State of Bavaria under the Excellence Strategy of the Federal Government and the Länder through the ONE MUNICH Project Munich Multiscale Biofabrication. P.T. acknowledges the support of BMBF (SIBOF, 03VP03891). D.C.L., P.T., and E.P. gratefully acknowledge the financial support of the Ludwig-Maximilians-Universität München via the Department of Chemistry, the Center for NanoScience (CeNS) and the LMUinnovativ program BioImaging Network (BIN).

## Author contributions

S.W. developed and implemented the deep-learning algorithm Deep-LASI and performed the Deep-learning assisted analyses, J.B. prepared the DNA origami samples under the supervision of P.T., P.A. collected the single-molecule TIRF data, P.A. and E.P. performed the manual analysis of the smFRET data, C.B.S. wrote the manual analysis software in which Deep-LASI was incorporated, S.W. and E.P. wrote the first draft of the manuscript, S.W. and E.P. designed the figures, all authors contributed to revising the manuscript. E.P. and D.C.L. supervised the project.

## Funding

## Competing interests

All authors (S.W., P.A., J.B., C.B.S., P.T., E.P., and D.C.L.) declare no competing interests.
