## [Peer Review File · Nature Communications]

REVIEWERS' COMMENTS

Reviewer #1 (Remarks to the Author)

The authors present a software package based on deep-neural networks (DNN) to analyze two and three colour smFRET data. The algorithm is tested against simulations and experimental data that has been manually analysed.

Analysis of smFRET data requires painstaking trajectory sorting and advanced statistical analysis (HMM). The use of DNN dramatically speeds up and facilitates analysis of smFRET trajectories.

The manuscript is well organized, the claims are properly supported in the SI and several examples of the algorithm performance and capabilities are shown.

Before publication, the following comments should be addressed.

1. The results in the Further applications section are not well explained. The expected FRET behaviour of the different DNA origami constructions (7, 6.5, 6 and 5 nt overhangs) should be further explained. Not everybody is as familiar with DNA origami constructs as the authors.
2. The sg-FCS reference requires to be updated
3. A one-sentence explanation of sgFCS would help, so that the reader does not need to download the sgFCS paper to know what the authors are referring to.
4. The claim that a 3°C is enough to explain the different dwell times obtained using a TIRF and a confocal configuration is plausible, but not proven. TIRF and confocal acquisition are different (temporal resolution, ms-long readout time in TIRF configuration, etc) This claim should be validated (measuring at the same T with both techniques, for instance) or the sentence qualified.
5. The github link for the code is missing
6. The print quality of all figures must be improved. Some fonts are too small, colours in panels containing FRET trajectories are way too similar (light and dark green, for instance). Fig. SN5.1 is almost unreadable, etc.

Reviewer #2 (Remarks to the Author)

Summary Statement

In the manuscript entitled "Deep learning assisted single molecule imaging analysis (Deep-LASI) of multi-color DNA origami structures," the authors use machine learning tools to accelerate and reduce biases in FRET analysis. The authors use an L-shaped DNA origami to test their Deep-LASI algorithm with increased complexity. Authors compare their results to results obtained by standard protocols to validate their approach.

After reading the manuscript, here are the noteworthy results:

- The results presented show that Deep-LASI has the capability to improve the reproducibility of FRET analysis. This topic is of current interest to the FRET community (Götz, M., Barth, A., Bohr, S.SR., et al. A blind benchmark of analysis tools to infer kinetic rate constants from single-molecule FRET trajectories. Nat Commun. 13, 5402 (2022).)
- Deep-LASI seems to simplify how to best determine the calibration parameters for precision FRET experiments (Hellenkamp, B., Schmid, S., Doroshenko, O. et al. Precision and accuracy of single-molecule FRET measurements—a multi-laboratory benchmark study. Nat Methods 15, 669–676 (2018)).
- Deep-LASI also shows the potential to reduce the challenges associated with multi-color FRET analysis. Multi-color FRET has yet to be used in rather complex systems, and likely, Deep-LASI could aid in the complexities of analyzing such data.

However, the manuscript also contains some significant weaknesses:

- Machine Learning analysis heavily depends on its training dataset. When using modeled data to

train ML, there is an intrinsic bias by the one setting the training dataset. A comparison of using different training data sets could aid in mitigating these weaknesses.

- Once the data set is trained, Deep-LASI can be first tested against the published dataset and benchmarked study (Götz et al 2022) as a first step before getting into their L-shape origami scaffolds.
- Considering that the training consists of synthetic data and its effects on who trains the DL algorithm, an analysis of S/N is lacking. In other words, what are the limits of Deep-LASI? Except for the 3-color FRET, most of the Origami-based data sets seem clean and optimal. What would happen when using more noisy data, and what is the success rate and accuracy?

In addition, other details could improve the readability and aid in reproducibility.

- This manuscript uses an L-Shape Origami scaffold for their validation. I think that it is important to mention other characterizations of these scaffolds. Although some of that work is published, it could prevent going back and forth between different manuscripts to gather those details. These are important in contextualizing the observed results.
- On a similar note, Deep-LASI is compared to manual selection providing similar but not identical numbers. Was there any effort to go back and understand events that were selected by one but not the other and vice versa? How many of those events were present? Also, what about all the non-selected events? Are those due to incomplete scaffolds or other experimental limitations? Could authors elaborate on those?
- Multiple details of parameters used in the CNN lack justification, including details of the implemented loss function and default parameters. Maybe these will be provided with the code, but it is important to set parameters and provide a rational justification without assuming that the code will be properly documented. This will improve reproducibility when people get access to the code.
- In the paragraph, the authors write "appropriate data sets". Can authors elaborate on what entails an appropriate data set, or better yet, provide an example of bad data set and results with bad training? Will there be false positives if trained wrongly?
- On page 2, paragraph 2, the purpose of enumerating the obstacles of quantitative smFRET rather than just mentioning them is unclear. I suggest the authors explain how these restrictions could affect Deep-LASI.
- On page 5, the first paragraph of the section "Training of Deep-LASI", when the authors mention "all possible FRET," it is understood that your software covers all the infinity number of FRET combinations, which cannot be possible because the training data is finite. Could the authors be more explicit on what they mean?
- On page 7, first paragraph of section "Single-Color experiments", the distances between the binding sites and the pointer position and between the quencher and the binding site are not justified.
- Add "panels" in the caption of Figure 3 to complement the idea that (a-e) and (f-j) refer to the panels.

Reviewer #3 (Remarks to the Author)

****Key results****

This paper introduces a new deep neural network based framework for the analysis of single molecule FRET (smFRET) experiments. The authors propose their new method as a replacement for time consuming and human bias prone existing smFRET data analysis method such hidden Markov models (HMMs). They evaluated their methods with an exhausting set of experiments on real data, and show that Deep-LASI outperforms HMM based methods.

****Expertise****

The scope of my expertise in the review process of this manuscript encompasses the design and training of the hidden Markov models.

****Significance****

The significance of the conclusions for the field of smFRET data analysis is outside of the scope of

my expertise for this manuscript.

****Validity****

The evaluation methods are overall sound and well described in the paper.

My main concern regarding the validity of the comparison of Deep-LASI with HMMs is the way both models were trained. Deep-LASI can learn from the global dataset while HMMs are restricted to learn from each trace individually. The rationale for this is given in S26: "We used a trace-wise approach when analyzing 1-color and apparent FRET traces as the exact value of the state can be shifted due to the above-mentioned background contributions.". Being unfamiliar with the practices in the field I am asking the authors to clarify this choice and comment on the use of more advanced emission distributions which could allow HMMs to be trained globally while not suffering from background contributions to the measurements. (More details below)

****References****

The references related to HMMs are adequate.

****Hidden Markov Modeling****

****Model definition****

- (Eq. 5.17) I am not familiar with the definition of the likelihood with exponents of the posterior. I think that the likelihood that the authors introduce here is not the likelihood per say but the proxy optimization criterion that is used to learn the HMM parameters in the expectation-maximization (EM) framework. For clarity I recommend to reformulate the HMM model introduction as follows: (1) introduce the factorization of the joint distribution $p(x_1, \dots, x_T, q_1, \dots, q_T)$, (2) introduce the EM optimization criterion and (3) introduce the iterative parameter update rules for the state transition kernels and the emission distribution parameters.

- (p5) "The summed probabilities over all unbleached frames serve as confidence levels for each trace." HMM have the abilities to return posterior state probabilities resulting from explicit modeling choices made for the emission distributions. Here the authors replace those by interpreting the output of DNNs as posterior probabilities. Please comment on the potential pitfalls/advantages of exploiting "artificial" confidence intervals versus using some emerging from explicit modeling choices ?

- The authors use Gaussian emission distributions in their HMM. This is a very restrictive choice since it does not allow multi-modal state emission distributions. More advanced models such as Gaussian Mixture Models could be used to address potential batch effects for instance. The author seem to acknowledge such issues (S26: "We used a trace-wise approach when analyzing 1-color and apparent FRET traces as the exact value of the state can be shifted due to the above-mentioned background contributions."), and choose to train HMMs per-trace rather than globally. Please comment on the use of more advanced emission distributions in this context instead of Gaussian emission.

In my opinion, training both models globally would allow for a fairer comparison of the modeling abilities of HMM vs Deep-LASI. any comments regarding this ?

- (S5) "The signal and noise characteristics of smFRET data is well enough understood that simulated data can accurately reproduce the characteristics of real data." Please comment on simulation model used, and specifically whether the HMM emission distribution design is in line with the knowledge employed to synthesize training data for Deep-LASI ?

- HMM are generative models that allow the sampling of random sequences from a trained model.
- The generation of random sequences is a way to provide insights regarding what was learnt during model training, and provides a sanity check mechanism for the model. This is not possible using the ensemble DNN method employed by the authors since trained DNNs/CNN-LSTM remain difficult to investigate. Please comment on the importance of model interpretation/sanity check in this context.

- The generation of sequences could also be leveraged as a data augmentation technique to further train DNN/CNN-LSTM models. Please comment on the use of this method to complement the current synthetic data generation method.

****Model training****

- (Eq. 5.15) The posterior seem to be recursively defined since the same variable appears on both sides of the equality. The HMM model exposition would be clearer if the authors describe the training algorithm used to learn the parameters.

- (S26) The transition matrices and the emission distributions are learnt per traces rather than globally for all traces. This is not the case for the Deep-LASI that can learn from all the traces. Please comment on how this impacts the HMM model's performances.
- (S26) Similarly, Deep-LASI is trained on synthetic data as well as real data but the HMMs are trained only on real data. Please comment on whether using synthetic data to train the HMMs would have an impact on the performances of HMM.

****Model evaluation****

- (Figure SN3.5) The results of the comparison of Deep-LASI with HMM is clearly presented and convincing. However my expertise is limited regarding the adequacy of these comparisons with the claims of the authors.

****Clarity and context****

- (Intro p2): "Low statistics" Please clarify this terminology
- (Intro p2) : "Moreover, it assumes that a target state is dependent only on the current state." I am not sure what a target state means here, I suggest changing to "next state"
- I am confused by the use of "trajectories", "traces", "sections" and "frames" through out the manuscript. I am assuming that "trajectories" and "traces" refer to the same thing: the entire time series; and the "sections" and "frames" also refer to the same thing: some portions of the traces. Please clarify these terms, and describe the choice of the length of the frames/sections

****Minor****

- (Eq. 5.12) The denominateur should be $\sqrt{2\pi}\sigma_{qi}$, not $\sqrt{2\pi}\sigma_{qi}^2$, for a correctly scaled density
- (Eq. 5.16) Some issues in the choice of the indexes

Reviewer #4 (Remarks to the Author)

The article discusses the development of Deep-LASI, a software package that uses deep neural networks to analyze single-molecule data, specifically from smFRET experiments rapidly. This software package readily applies to as many as three molecular interactions, as demonstrated using tethered DNA molecules on DNA nanostructures. While the software offers faster processing times and reduced human bias, several limitations exist.

- The article lacks a discussion on the potential limitations of using deep neural networks in single-molecule data analysis. While deep neural networks have shown promising results in various applications, there is a risk of overfitting, leading to inaccurate predictions when the model is tested on new data. Therefore, it is important to understand deep neural networks' potential drawbacks and limitations in the context of single-molecule data analysis.
- The authors only benchmarked Deep-LASI against a conventional Hidden Markov Model analysis. Still, they did not compare it against other commonly used software packages, which would be important to determine its relative performance. There are many analysis software packages published in the past. But analysis of smFRET is more routinely done by manual approach than automated software packages because of the inherent unpredictable complexity of smFRET data. Hopefully, Deep-LASI will stand out and will be adopted by the community. In this regard, authors should demonstrate the application of Deep-LASI in more realistic conditions where background noise is higher. For instance, most single-molecule FRET experiments are done with donor-labeled molecules floating in solution and acceptor-labeled molecules immobilized on the surface. This will substantiate their claims and show the wider application of the analysis technique.
- Any specific reason for choosing L-shape origami? Would a plane rectangular structure work for the same?
- Kinetic scheme showing individual states and transitions between them would be great for quickly grasping the single-, two-color, and three-color FRET experiments on L-shaped origami.
- Does Deep-LASI need to be trained with data containing FRET states that are expected to be detected? In that case, the user has to define the states in the training data set?
- In FRET experiments, Cy3B was attached to a flexible linker. What's the linker length for Cy3B and the other two protrusions? It is not clear from the figures presented.

- In the single-color data, should there not be three states? Bound-quenched (state 1), bound-unquenched (state2), and unbound state (not defined in the scheme). A control experiment without the state-1 and state-2 protrusions would reveal the undefined state.
- Why are the binding times of state-1 and state-2 for single-color data are substantially different given that they are composed of the same sequence? Also, for a 7.5nt complementary strands hybridization, the measured binding time is longer than previously reported (<https://pubmed.ncbi.nlm.nih.gov/20957983/> and <https://pubmed.ncbi.nlm.nih.gov/32787168/>). Especially that they performed their experiments in low salt buffer (only 50 mM KCl and 100 Tris-HCl (pH?)), the expected binding times are lower than the reported ones. An explanation and discussion on this are required.
- A table showing all the binding times for each case is required as it is difficult to read them from figures.
- Authors observe decrease in binding times upon introducing glycerol in the imaging buffer. They explain the difference by taking the hydration shell into account. Another factor that is more interesting in my opinion is to consider the slow down of transitions enabling to clearly
- Page 3, second paragraph, grammatical error in ‘...for determine of FRET correction..’ should be ‘...for determining FRET correction...’
- ‘...the pretrained neural networks operate locally on each trace, they do not neglect rare events which would be missed in global analysis approaches and lead to inaccurate conclusions.’ It would be great to demonstrate the same for substantiating the claim.
- Correction in FRET: it is unclear what exactly is corrected.
- In the following sentence, ‘Considering the camera exposure times of 50 ms (6.5 nt, 7 nt and 7.5 nt data sets) and 150 ms (8 nt data set), a dwell-time to frame-time ratio ranges from 10 (6.5 nt State 1) to 62 (8 nt, State 2).’, isn’t the dwell-time to frame ratio 2 (100 msec/50msec) for (6.5 nt State 1) and 22 (3300 msec/150msec) for (8nt state 2)?
- What’s the pH of photostabilization buffer? Surprisingly, the buffer contains low ionic strength without divalent cations. In such buffer, DNA origami structures start disassembling.
- How long is the enzymatic oxygen depletion time?
- Please provide exact mono- and bi-exponential equations used for fitting.

RESPONSE TO REVIEWERS' COMMENTS

Reviewer #1 (Remarks to the Author)

The authors present a software package based on deep-neural networks (DNN) to analyze two and three colour smFRET data. The algorithm is tested against simulations and experimental data that has been manually analysed.

Analysis of smFRET data requires painstaking trajectory sorting and advanced statistical analysis (HMM). The use of DNN dramatically speeds up and facilitates analysis of smFRET trajectories.

The manuscript is well organized, the claims are properly supported in the SI and several examples of the algorithm performance and capabilities are shown.

We thank the reviewer for her/his overall positive evaluation of our manuscript.

Before publication, the following comments should be addressed.

1. The results in the Further applications section are not well explained. The expected FRET behaviour of the different DNA origami constructions (7, 6.5, 6 and 5 nt overhangs) should be further explained. Not everybody is as familiar with DNA origami constructs as the authors.

We apologize for this oversight. We now introduce the expected behaviors for the different DNA origamis in the further application section as follows; "We chose the same double-labeled DNA origami structure introduced in Figure 2a, but with different combinations of docking strands. For these constructs, the measured FRET efficiencies will be the same but with different dynamics. By changing the hybridization length or by adding mismatched bases in the docking strand, the interaction time of each binding site can be tuned individually from ~ 1 ms to 10 s of seconds by adjusting the stabilization energy of DNA hybridization. As expected, shorter hybridization sequences lead to fast dynamics."

2. The sg-FCS reference requires to be updated

We have updated the sgFCS reference.

3. A one-sentence explanation of sgFCS would help, so that the reader does not need to download the sgFCS paper to know what the authors are referring to.

We have now added a short explanation of sgFCS in the revised manuscript: "In sg-FCS, a pulsed light source is used such that the fluorescence lifetime information can be included into the analysis. By shrinking the analysis window of photons based on their detected arrival time after excitation, we vary the relative brightness of two species with different fluorescence lifetimes (e.g. the low FRET and high FRET states). This leads to a robust extraction of the kinetic rates between the two states from the autocorrelation analysis of the FCS data."

4. The claim that a 3°C is enough to explain the different dwell times obtained using a TIRF and a confocal configuration is plausible, but not proven. TIRF and confocal acquisition are different (temporal resolution, ms-long readout time in TIRF configuration, etc) This claim should be validated (measuring at the same T with both techniques, for instance) or the sentence qualified.

We have now added measurements of the two-color, two-state L-shaped DNA origami structure at four different temperatures. A change in 2°C leads to roughly a factor 2 change in the rates. This data has been added to Supplementary Materials as Supplementary Note 6.4.

5. The github link for the code is missing

The github link and documentation are now given in the revised manuscript: "A version of the program is available on GitHub [<https://gitlab.com/simon71/deeplasi>]. Extensive documentation for the software can be found at <https://deep-lasi-tutorial.readthedocs.io/en/latest/index.html>."

6. The print quality of all figures must be improved. Some fonts are too small, colours in panels containing FRET trajectories are way too similar (light and dark green, for instance). Fig. SN5.1 is almost unreadable, etc.

We thank the reviewer for pointing this out. This was partially due to the compressed version of the figure sent out for review. We have now updated the font-sizes (within the specifications of the Nature publishing group), color and resolution of the images. We have also reconstructed figure SN5.1. We hope, with the improved resolution, font size and colors are more easily distinguishable.

Reviewer #2 (Remarks to the Author)

Summary Statement

In the manuscript entitled "Deep learning assisted single molecule imaging analysis (Deep-LASI) of multi-color DNA origami structures," the authors use machine learning tools to accelerate and reduce biases in FRET analysis. The authors use an L-shaped DNA origami to test their Deep-LASI algorithm with increased complexity. Authors compare their results to results obtained by standard protocols to validate their approach.

After reading the manuscript, here are the noteworthy results:

- The results presented show that Deep-LASI has the capability to improve the reproducibility of FRET analysis. This topic is of current interest to the FRET community (Götz, M., Barth, A., Bohr, S.SR., et al. A blind benchmark of analysis tools to infer kinetic rate constants from single-molecule FRET trajectories. Nat Commun. 13, 5402 (2022).)
- Deep-LASI seems to simplify how to best determine the calibration parameters for precision FRET experiments (Hellenkamp, B., Schmid, S., Doroshenko, O. et al.

Precision and accuracy of single-molecule FRET measurements—a multi-laboratory benchmark study. Nat Methods 15, 669–676 (2018)).

- Deep-LASI also shows the potential to reduce the challenges associated with multi-color FRET analysis. Multi-color FRET has yet to be used in rather complex systems, and likely, Deep-LASI could aid in the complexities of analyzing such data.

We thank the reviewer for her/his appreciation of the advances of our introduced analysis approach. We would like to mention that we now also demonstrate the use of Deep-LASI using the data from the Hellenkamp et al paper in the software documentation found at: <https://deep-lasi-tutorial.readthedocs.io/en/latest/starter.html>.

However, the manuscript also contains some significant weaknesses:

- Machine Learning analysis heavily depends on its training dataset. When using modeled data to train ML, there is an intrinsic bias by the one setting the training dataset. A comparison of using different training data sets could aid in mitigating these weaknesses.

For Deep-LASI, we have used an extensive simulated data set with ~ 200,000 traces for training each network covering an extensive range of realistic experimental parameters to avoid any bias in the analysis. This includes FRET efficiencies between 0.01 and 0.99, dwell times of 1 to 100 frames and SNR of ~ 0.3 to 50. To test how robust the training data set is, we have generated two new training sets and compared the impact of the training datasets on the analysis of two-color, two-state DNA origami data. The differences between using different training sets were less than that between individual users. This information has now been added to the results section for the DNA origami data and described in detail in the new Supplementary Note 4.2.

In the results section on *Dual-Color Experiments*, we have added the following paragraph: "Next, we investigated how sensitive Deep-LASI is to the training dataset. Hence, we trained two additional classifier networks using newly simulated datasets. Details are given in Supplementary Note 4.2. The consistency between the differently trained neural networks is ~ 90 %, similar to what would be expected from analysis run on the validation datasets (Supplementary Figure SN3.1c). Interestingly, the consistency between the different neural networks is higher than between two independent users (Supplementary Figure SN4.3b)."

- Once the data set is trained, Deep-LASI can be first tested against the published dataset and benchmarked study (Götz et al 2022) as a first step before getting into their L-shape origami scaffolds.

As requested by the reviewer, we have now moved our comparison with the benchmarked study from the end of the internal benchmarking with DNA origamis to the end of the section on the **performance of Deep-LASI**. We have also moved the SI discussion of the comparison from Supplementary Note 6.4 to Supplementary Note 3.7.

- Considering that the training consists of synthetic data and its effects on who trains the DL algorithm, an analysis of S/N is lacking. In other words, what are the limits of Deep-LASI? Except for the 3-color FRET, most of the Origami-based data sets seem clean and optimal. What would happen when using more noisy data, and what is the success rate and accuracy?

We apologize for not being more explicit with respect to the SNR. In section SN3.4, we now remind the reader that the signal intensity is normalized to 1 for the detailed noise analysis give in Figure SN3.5. Hence, the value of the noise is $1/\text{SNR}$. In addition, the data from Hsp70 shown in Figure 6c-f have a lower signal-to-noise as photostabilizers could not be used during these measurements. We now compare example traces that were selected manually and/or by Deep-LASI and discuss the results in the new Supplementary Note 6.5. We have added the following information to the paragraph on smFRET with proteins: "As the proteins were immobilized by incorporation in vesicles, a photostabilization buffer could not be used for the experiments. Hence, the signal-to-noise ratio in these experiments are lower than those exhibited by the photostabilized DNA origami structures. A comparison of traces selected manually and/or by the Deep-LASI analysis are discussed in Supplementary Note 6.5."

The new Supplementary Note 6.5 reads as: "To test Deep-LASI on single-molecule FRET data from proteins, we reanalyzed data that we published previously¹⁴. These data were collected of proteins that were encapsulated in ~200 nm liposomes. Due to vesicle encapsulation, a photostabilization buffer could not be used. Hence, the protein data on this system had a lower signal-to-noise ratio than we typically had with the photostabilized DNA origami structures. In Supplementary Figure SN6.5, we show examples of individual traces that were evaluated similarly or differently by the user and Deep-LASI. From these comparisons, we see that: 1) Deep-LASI tends to take shorter regions of traces that are trashed in the manual analysis, 2) that manually selected dynamic traces with high noise tend to be categorized by Deep-LASI as noisy rather than dynamic, and 3) traces where the dyes are initially off at the beginning of the trace due to photophysics are typically categorized as artifacts in Deep-LASI whereas a user may manually select the middle region of the trace."

In addition, other details could improve the readability and aid in reproducibility.

- This manuscript uses an L-Shape Origami scaffold for their validation. I think that it is important to mention other characterizations of these scaffolds. Although some of that work is published, it could prevent going back and forth between different manuscripts to gather those details. These are important in contextualizing the observed results.

We chose the L-Shape origami structure that has been previously characterized and published (Kamińska et al. *Advanced Materials* 2021 **33**: 2101099; Schroder et al *Proc Natl Acad Sci U S A* 2023 **120**: e2211896120; Krause et al. *ACS Nano* 2021 **15**: 6430-6438; Masullo et al. *Nano Lett* 2021 **21**: 840-846.) as it is an ideal test sample for our software. We introduce the L-shaped origami in more detail at the beginning of the results section of **Deep-LASI analyses of DNA origami structures**: "The geometry of the DNA

structure was originally designed for measuring energy transfer to a graphene surface^{5,30}. The single-stranded DNA pointer along with two or three exchangeable docking strands of different complementary sequences allows the number of states, position of the dyes, and kinetic rate to be programmed as desired. Hence, it is an ideal test system for measuring and extracting kinetic information from smFRET traces." We have also added AFM and TEM images of the structure in Supplementary Figure SN7.2.

- On a similar note, Deep-LASI is compared to manual selection providing similar but not identical numbers. Was there any effort to go back and understand events that were selected by one but not the other and vice versa? How many of those events were present? Also, what about all the non-selected events? Are those due to incomplete scaffolds or other experimental limitations? Could authors elaborate on those?

We thank the reviewer for her/his suggestion. We now provided a more detailed comparison between the manual and Deep-LASI analysis in Supplementary Note 4. The main sources for the discrepancies are: 1) The categorization of short traces. Deep-LASI can include shorter regions of traces that are trashed in the manual analysis or, depending on the context of the trace, discard short traces due to non-ideal donor and/or acceptor signals that would be manually selected. 2) Fast dynamic traces with high noise. Deep-LASI tends to categorize fast dynamics or traces with high noise as noisy, even if it is only in a short region of the trace. Users may select the trace or none-noisy regions of the trace. These types of traces can be included by lowering the confidence level for inclusion of the traces. 3) Traces where the dyes are initially off at the beginning due to photophysics. Middle regions may be selected manually but tend to be categorized as artifacts in Deep-LASI.

Regarding the number of events, this value varies significantly depending on the user. When analyzing the 2-color DNA origami from Figure 3, one user tried to maximize statistics, also selecting short regions of traces, leading to a total of ~1700 dynamic traces in total whereas a second user only selected the ideal traces, limiting the total dynamic traces to ~ 1000. This information is now included in Supplementary Figure SN4.3b.

The non-selected events can have many origins including but not limited to 1) incomplete labeling, 2) quick photobleaching, 3) a high amount of photophysics (e.g. blinking), especially near the end of the lifetime of the photostabilization buffer and 4) other molecules in the vicinity that effect intensity or background.

- Multiple details of parameters used in the CNN lack justification, including details of the implemented loss function and default parameters. Maybe these will be provided with the code, but it is important to set parameters and provide a rational justification without assuming that the code will be properly documented. This will improve reproducibility when people get access to the code.

In Supplementary Note 1.1, we now give more details regarding our choice for the network and related parameters (see highlighted changes). The code is available under gitlab (<https://gitlab.com/simon71/deeplasi>) and is extensively documented <https://deep-lasi-tutorial.readthedocs.io/en/latest/starter.html>.

- In the paragraph, the authors write “appropriate data sets”. Can authors elaborate on what entails an appropriate data set, or better yet, provide an example of bad data set and results with bad training? Will there be false positives if trained wrongly?

We apologize for the unclarity. Here, with "appropriate data sets", we mean data types for the corresponding classifier model (one color data, two-color data without ALEX, two-color data with ALEX and three-color data). We have rewritten the text to read: "To use Deep-LASI for analyzing single molecule data, we first trained the trace-classifier network on datasets appropriate for the corresponding network (i.e. one-color data, two-color data without ALEX, two-color data with ALEX or three-color data)."

Beyond clarification of the sentence, the reviewer requests looking into results when training with a "bad data set". In deep learning, "bad training" refers to 1) using a training dataset that is too small in size, 2) using a training dataset where there is intrinsic bias or 3) overfitting the given training dataset. It is obvious that a "bad training" dataset will give undesirable results including false positives. Garbage in - Garbage out. Currently, the code is written to ensure that the optimal training dataset for the given parameters is generated. We discuss what is important in selecting a training dataset at the beginning of Supplementary Note 2. The paragraph reads: "Although an optimized architecture is important and improves the efficiency of a neural network, the functionality of the network rises and falls with the dataset used for training the network. There are a number of important factors to consider when training a neural network. Typical pitfalls include using a dataset that is too small in size or contains an intrinsic bias, or overfitting the training data. A neural network is biased towards features it has seen before. Hence, the training dataset should include the various possibilities (e.g. number of FRET states, kinetic rates, signal-to-noise ratios). If the training dataset includes any bias, this will also be reflected in the output of the algorithm. One way that bias can be introduced into the training dataset is from unbalanced sampling of categories. For example, for the trace classifier models, it is important that the training dataset includes the same number of traces from each category. It is also important to know when to stop the training process. Neural networks can be overtrained, meaning that they memorize the training data but do not learn the general principles behind it. Below we discuss the details of the training procedure and how we optimized the training process."

- On page 2, paragraph 2, the purpose of enumerating the obstacles of quantitative smFRET rather than just mentioning them is unclear. I suggest the authors explain how these restrictions could affect Deep-LASI.

We have removed the enumeration in the sentence. The main point here is that, the more complex the system is, the more likely it is that the number of usable traces per movie decreases. This impacts the single molecule data analysis in general regardless of whether the data are manually analyzed or analyzed with Deep-LASI. With low quality data, Deep-LASI will classify most traces as noisy, aggregate or artifact and only a few are left for further analysis. This can be adjusted by lowering the required confidence levels for classifying the data as static or dynamic. We have changed the text to read: "Quantitative smFRET data analysis is strongly hampered by experimental restrictions

due to, for example, a low number of usable single molecule traces, data with a low signal-to-noise ratio (SNR), or short traces due to photochemistry. Overcoming these limitations requires large data volumes as very few molecules contain the desired information with suitable quality, which significantly increases the efforts involved in sorting through the data when performed manually."

- On page 5, the first paragraph of the section "Training of Deep-LASI", when the authors mention "all possible FRET," it is understood that your software covers all the infinity number of FRET combinations, which cannot be possible because the training data is finite. Could the authors be more explicit on what they mean?

What we implied is all possible FRET efficiencies (within reason) considering the number of states and noise levels of the system have been covered. We have removed "all possible" and the sentence now reads: "The training datasets were designed to cover a wide range of experimental conditions and FRET efficiencies." The details are giving in Supplementary Note 2.3. We simulate quenching levels for one color and FRET states in two and three colors randomly selected between 0.01 and 0.99, dwell time to exposure ratios of 1 to 100 and noise levels (or SNR) from ca 0.02 to ~ 3 (0.3 to 50).

- On page 7, first paragraph of section "Single-Color experiments", the distances between the binding sites and the pointer position and between the quencher and the binding site are not justified.

Here, we have used the L-shaped origami that has been extensively investigated by the group of Philip Tinnefeld. It was ideal for our purposes in that it provided visibly distinguishable states and adjustable kinetics. In the one-color case, we used the same structure with the acceptor as the quencher where the fluorescence from the acceptor channel was ignored rather than using a black-hole quencher. In this way, we did not have to introduce yet another structure/system into the paper. For the FRET states, we needed a significant difference in FRET efficiency between the two binding sites with measurable FRET values. We also wanted to avoid the extreme FRET values where, for example, at high FRET values, Dexter transfer can occur and/or the dyes interact. Due to the limited number of available positions with the correct orientation of the double helix extruding from the surface of the DNA origami structure, these were the most reasonable choices. We now justify our choice of the origami structure when we introduce the construct in the results section regarding **Deep-LASI analyses of DNA origami structures**: "The geometry of the DNA structure was originally designed for measuring energy transfer to a graphene surface^{5,30}. The single-stranded DNA pointer along with two or three exchangeable docking strands of different complementary sequences allows the number of states, position of the dyes, and kinetic rate to be programmed as desired. Hence, it is an ideal test system for measuring and extracting kinetic information from smFRET traces. "

- Add "panels" in the caption of Figure 3 to complement the idea that (a-e) and (f-j) refer to the panels.

We have added "panels" to the figure caption as requested.

Reviewer #3 (Remarks to the Author)

****Key results****

This paper introduces a new deep neural network based framework for the analysis of single molecule FRET (smFRET) experiments. The authors propose their new method as a replacement for time consuming and human bias prone existing smFRET data analysis method such hidden Markov models (HMMs). They evaluated their methods with an exhausting set of experiments on real data, and show that Deep-LASI outperforms HMM based methods.

****Expertise****

The scope of my expertise in the review process of this manuscript encompasses the design and training of the hidden Markov models.

****Significance****

The significance of the conclusions for the field of smFRET data analysis is outside of the scope of my expertise for this manuscript.

****Validity****

The evaluation methods are overall sound and well described in the paper.

My main concern regarding the validity of the comparison of Deep-LASI with HMMs is the way both models were trained. Deep-LASI can learn from the global dataset while HMMs are restricted to learn from each trace individually. The rationale for this is given in S26: "We used a trace-wise approach when analyzing 1-color and apparent FRET traces as the exact value of the state can be shifted due to the above-mentioned background contributions.". Being unfamiliar with the practices in the field I am asking the authors to clarify this choice and comment on the use of more advanced emission distributions which could allow HMMs to be trained globally while not suffering from background contributions to the measurements. (More details below)

We thank the reviewer for her/his insights into the comparison with HMM.

****References****

The references related to HMMs are adequate.

****Hidden Markov Modeling****

****Model definition****

- (Eq. 5.17) I am not familiar with the definition of the likelihood with exponents of the posterior. I think that the likelihood that the authors introduce here is not the likelihood per say but the proxy optimization criterion that is used to learn the HMM parameters in the expectation-maximization (EM) framework. For clarity I recommend to reformulate the HMM model introduction as follows: (1) introduce the factorization of the joint distribution $p(x_1, \dots, x_T, q_1, \dots, q_T)$, (2) introduce the EM optimization criterion and (3) introduce the

iterative parameter update rules for the state transition kernels and the emission distribution parameters.

We apologize for the confusion. What we refer to here is the loglikelihood function, which is the estimation maximization function. The definition follows along the line of (Zarrabi et al, 2018). We have now rewritten the HMM section Supplementary Note SN5.6.

- (p5) "The summed probabilities over all unbleached frames serve as confidence levels for each trace." HMM have the abilities to return posterior state probabilities resulting from explicit modeling choices made for the emission distributions. Here the authors replace those by interpreting the output of DNNs as posterior probabilities. Please comment on the potential pitfalls/advantages of exploiting "artificial" confidence intervals versus using some emerging from explicit modeling choices ?

We have now added a discussion section to the paper discussing the advantages and potential pitfalls of DNNs.

- The authors use Gaussian emission distributions in their HMM. This is a very restrictive choice since it does not allow multi-modal state emission distributions. More advanced models such as Gaussian Mixture Models could be used to address potential batch effects for instance. The author seem to acknowledge such issues (S26:"We used a trace-wise approach when analyzing 1-color and apparent FRET traces as the exact value of the state can be shifted due to the above-mentioned background contributions."), and choose to train HMMs per-trace rather than globally. Please comment on the use of more advanced emission distributions in this context instead of Gaussian emission.

The choice of the emission function is determined by the noise characteristics of smFRET data. For each individual FRET state, this distribution is very well represented by a Gaussian function. While photon detection is typically Poissonian in nature, the Poisson distribution can be well approximated by a Gaussian distribution for average values above ca 10 (Martin, Statistics for Physicists, 1971). As we have shown, the FRET efficiency can also be approximated by a Gaussian distribution (Zarrabi et al, Biophysics J, 2018, see Supplementary Information). Thus, a multimodal distribution would be inappropriate for the single-state emission function. Also, the simulated data, which we used to compare Deep-LASI and HMM are generated using Gaussian noise. We also tested the use of multi-modal emission functions for the HMM analysis with no significant differences in the results. Of course, traces with multiple FRET states have different Gaussian emission functions based on the value of the FRET efficiency. We mention this now explicitly in our description of HMM, page S34: "For intensity measurements and single molecule FRET traces, it is appropriate to model the emission probability of a state q_i as a Gaussian distribution:^{12,13}".

In my opinion, training both models globally would allow for a fairer comparison of the modeling abilities of HMM vs Deep-LASI. any comments regarding this ?

We agree that the comparison was limited to a local analysis of smFRET traces and it does not seem fair if the comparison is solely based on the capabilities of removing background contributions. HMM assumes that the data can be reproduced by a single set of parameters while smFRET data typically contain heterogeneities. Comparing Deep-LASI and global HMM in a heat map is not feasible for us due to the amount of data and time needed for the analysis (ca 300 traces per data point for a total of ~ 300,000 data points). The same is true for 2-dimensional local HMM, which takes approximately 6-10 seconds per trace for training. We recognize that there is no reason to assume that deep neural networks can significantly outperform global HMM with the correct prior assumptions when the dataset is large, homogeneous and low in dimensionality. Therefore, we included the precision of global HMM and Deep-LASI of one dataset containing 2000 traces with time independent noise distributions to emphasize that the advantages of Deep-LASI may only be true in the field of single molecule data analysis. In this comparison, the performance increase is clearly attributed to the higher dimensionality of the data, which is a strong suit of deep neural networks. We have now included a comparison of local and global HMM analyses with Deep-LASI in the new Supplementary Note 3.5.

Lastly, we would like to note the key differences in the training of Deep-LASI and global HMM. As the reviewer correctly points out, HMM is a generative model and learns the underlying set of parameters given the observed data. Deep-LASI is a discriminative model that is trained to only extract features from the observed data. For the state classifiers, these features are state transitions in the context of local noise without assuming mono-exponentially decaying dwell-time distributions. The training dataset for Deep-LASI is not compatible with a training dataset for HMM since a unique set of parameters is generated for every trace in the DNN training dataset, which contains the full range of sensible parameters for signal-to-noise, FRET states and kinetic rates.

- (S5) "The signal and noise characteristics of smFRET data is well enough understood that simulated data can accurately reproduce the characteristics of real data." Please comment on simulation model used, and specifically whether the HMM emission distribution design is in line with the knowledge employed to synthesize training data for Deep-LASI ?

Photon counting statistics are Poisson in nature. However, with average values above ca 10, the Poisson distribution can be well approximated by a Gaussian distribution (Martin, Statistics for Physicists, 1971). As we have shown, the FRET efficiency can also be approximated by a Gaussian distribution (Zarrabi et al, Biophysics J, 2018, see Supplementary Information). Hence, the simulated Gaussian noise corresponds precisely to the used HMM emission distributions.

- HMM are generative models that allow the sampling of random sequences from a trained model.

- The generation of random sequences is a way to provide insights regarding what was learnt during model training, and provides a sanity check mechanism for the model. This is not possible using the ensemble DNN method employed by the authors since trained

DNNs/CNN-LSTM remain difficult to investigate. Please comment on the importance of model interpretation/sanity check in this context.

We agree with the reviewer that it is important to perform sanity checks on the trained DNN. In contrast to HMM, the DNN network is trained once on ~ 200,000 traces randomly varied over the selected parameter range. The quality of this training was investigated first using additional, simulated data and then compared with human analysis of real data. The heat plots shown in Figure SN3.5 were generated using 300,000 traces that the network had not seen before covering an extensive range in noise, state separation, trace length and kinetics rates. This gives a good indication under which circumstances the DNN works reasonably well.

- The generation of sequences could also be leveraged as a data augmentation technique to further train DNN/CNN-LSTM models. Please comment on the use of this method to complement the current synthetic data generation method.

We are sorry for the misunderstanding. This is exactly what we do. We used an HMM routine (Schreiber et al) for generating the training dataset. We now mention this at the beginning of Supplementary Note SN2.3.

****Model training****

- (Eq. 5.15) The posterior seem to be recursively defined since the same variable appears on both sides of the equality. The HMM model exposition would be clearer if the authors describe the training algorithm used to learn the parameters.

Eq 5.15 and 5.16 are indeed recursive and the training is done reiteratively until the values converge. For the analysis, we used the Baum-Welch algorithm. We have rewritten the HMM description as mentioned above in Supplementary Note SN5.6. We have also moved the description of the HMM parameters from previous Supplementary Note 3.7 to the New Supplementary Note 5.7.

- (S26) The transition matrices and the emission distributions are learnt per traces rather than globally for all traces. This is not the case for the Deep-LASI that can learn from all the traces. Please comment on how this impacts the HMM model's performances.

We now also compared Deep-LASI with a local and global HMM analysis on simulated three-color, two-state data. One, two and three-color data from the simulation were analyzed and compared. The results are given in a new Supplementary Note SN3.5.

- (S26) Similarly, Deep-LASI is trained on synthetic data as well as real data but the HMMs are trained only on real data. Please comment on whether using synthetic data to train the HMMs would have an impact on the performances of HMM.

We apologize for the confusion. Deep-LASI is only trained on synthetic data. The HMM algorithm is trained on the dataset being analyzed, which can be either experimental or simulated. Certainly, the quality of the HMM results will depend on the quality of the

original data. A detailed discussion of the impact of analyzing synthetic data with HMM is summarized in Supplementary Note 3.5.

****Model evaluation****

- (Figure SN3.5) The results of the comparison of Deep-LASI with HMM is clearly presented and convincing. However my expertise is limited regarding the adequacy of these comparisons with the claims of the authors.

It is challenging to fairly compare HMM and DNN models over all possible types of data and training approaches. The analysis in supplementary Figure SN3.5 is our honest attempt to benchmark the various parameters important for the analysis of dynamic smFRET traces. In many cases, both HMM and DNN yield similar results. In addition, we have now also included a comparison with local and global HMM in Supplementary Note 3.5.

****Clarity and context****

- (Intro p2): "Low statistics" Please clarify this terminology

As mentioned in our response to reviewer 2, the main point here is that, the more complex the system is, the more likely it is that the number of usable traces per movie decreases. What defines "low statistics" depends on the details of the system and the question investigated. In the introduction, we wish to remain somewhat general rather than putting exact numbers of on the amount of usable single molecule traces or transitions researchers should use. We have rewritten the sentence to read: "Quantitative smFRET data analysis is strongly hampered by experimental restrictions due to, for example, a low number of usable single molecule traces, data with a low signal-to-noise ratio (SNR), or short traces due to photochemistry."

- (Intro p2) : "Moreover, it assumes that a target state is dependent only on the current state." I am not sure what a target state means here, I suggest changing to "next state"

We have changed the sentences as suggested by the reviewer. It now reads:

"Moreover, it assumes that the probability of a transition to the next state only depends on the current state."

- I am confused by the use of "trajectories", "traces", "sections" and "frames" through out the manuscript. I am assuming that "trajectories" and "traces" refer to the same thing: the entire time series; and the "sections" and "frames" also refer to the same thing: some portions of the traces. Please clarify these terms, and describe the choice of the length of the frames/sections

Trajectories and traces are terms that are used interchangeably in the field and refer to the entire time series for one molecule. Frames refer to the individual time points in a trace and sections refer to contiguous frames within a trace.

****Minor****

- (Eq. 5.12) The denominator should be $\sqrt{2\pi}\sigma_{qi}$, not $\sqrt{2}\pi\sigma_{qi}^2$, for a correctly scaled density

We thank the reviewer for pointing out this typo and have corrected it in the revised manuscript. The correct equation was used in the software.

- (Eq. 5.16) Some issues in the choice of the indexes

Equation 5.16 now reads:

$$w_{q_i,t} = \frac{W_q f_{q_i}(x_t | \mu_{q_i}, \sigma_{q_i})}{\sum_{i=1}^Q W_{q_i} f_{q_i}(x_t | \mu_{q_i}, \sigma_{q_i})}$$

Reviewer #4 (Remarks to the Author)

The article discusses the development of Deep-LASI, a software package that uses deep neural networks to analyze single-molecule data, specifically from smFRET experiments rapidly. This software package readily applies to as many as three molecular interactions, as demonstrated using tethered DNA molecules on DNA nanostructures. While the software offers faster processing times and reduced human bias, several limitations exist.

- The article lacks a discussion on the potential limitations of using deep neural networks in single-molecule data analysis. While deep neural networks have shown promising results in various applications, there is a risk of overfitting, leading to inaccurate predictions when the model is tested on new data. Therefore, it is important to understand deep neural networks' potential drawbacks and limitations in the context of single-molecule data analysis.

We have now added a discussion section where we discuss the advantages and pitfalls of deep-learning approaches to single molecule data analyses.

- The authors only benchmarked Deep-LASI against a conventional Hidden Markov Model analysis. Still, they did not compare it against other commonly used software packages, which would be important to determine its relative performance. There are many analysis software packages published in the past. But analysis of smFRET is more routinely done by manual approach than automated software packages because of the inherent unpredictable complexity of smFRET data. Hopefully, Deep-LASI will stand out and will be adopted by the community. In this regard, authors should demonstrate the application of Deep-LASI in more realistic conditions where background noise is higher. For instance, most single-molecule FRET experiments are done with donor-labeled molecules floating in solution and acceptor-labeled molecules immobilized on the surface. This will substantiate their claims and show the wider application of the analysis technique.

We agree with the reviewer that it is important to demonstrate the applicability of Deep-LASI to data with higher noise levels. Rather than adding another assay to this paper (i.e.

the single molecule binding assay suggested above), we show more details of the experiments with Hsp70 encapsulated in liposomes. Due to the encapsulation, oxygen scavenger systems and photostabilization approaches are not applicable. Hence, the signal quality of the data is significantly lower than for the DNA origami data. The Hsp70 data is discussed in Supplementary Note SN6.5, where we also discuss the similarities and difference between human analyses and Deep-LASI analysis. The analysis of DNA-paint data (binding, unbinding) will be included in a second manuscript.

While we indirectly compared Deep-LASI with other software packages used in the Kinsoft challenge (Supplementary Note 3.7), we also performed a comparison between DeepFRET and Deep-LASI. Below, are the confusion matrices for DeepFRET and Deep-LASI run on their own validation sets as well as on the validation set of the other software program I(Figure R1).

Figure R1: Confusion matrices showing (top) the performance of DeepFRET (left) and Deep-LASI (right) on their own data sets and (bottom) and the cross validation datasets.

Deep-LASI demonstrated a better performance on the cross-validation sets. We also analyzed two-color, two-state origami data with DeepFRET using the default parameters, which yielded very few dynamic traces. DeepFRET is more strict in selecting traces and thus more sensitive to noise. However, these are not fair comparisons for various reasons including the differences in the training datasets used. A more fair comparison would require retraining of each network on the same training dataset to compare the architectures. Hence, we decided not to include a comparison of the software in the manuscript.

- Any specific reason for choosing L-shape origami? Would a plane rectangular structure work for the same?

Our reason for using the L-shape origami was that it was already well characterized in previous publications and already available from the group of Philip Tinnefeld. It was initially made for experiments using graphene as an acceptor, hence the L-shape. In addition, the L-shaped origami platform is kinetically tunable and can be labeled with two or three colors. Hence, it is an ideal test case for the software. We have added this information in the beginning of the Results section regarding "**Deep-LASI analyses of DNA origami structures**". Of course, the analysis also would work on a planar structure.

- Kinetic scheme showing individual states and transitions between them would be great for quickly grasping the single-, two-color, and three-color FRET experiments on L-shaped origami.

We have added kinetic schematics to figures 2, 3, 4 and 5.

- Does Deep-LASI need to be trained with data containing FRET states that are expected to be detected? In that case, the user has to define the states in the training data set?

We have trained Deep-LASI over a wide range of FRET values (0.01 - 0.99), dwell time to exposure ratios (1 to 100) and SNR (ca 0.3 to 50). In contrast to HMM methods, no assumptions regarding the number of states, their FRET efficiencies and their widths need to be made. Deep-LASI has also been trained to recognize up to five different states and the kinetics can be analyzed in detail for up to four states. If more states are expected within a single trace, Deep-LASI would need to be retrained. How well the states are retrieved depends on the difference in FRET efficiency between the two states and the S/N ratio as discussed in Supplementary Figure SN3.5. We also added a comment in the main text in the section regarding **Training of Deep-LASI**: "Hence, no initial estimation of the number of states and expected FRET efficiencies are needed."

- In FRET experiments, Cy3B was attached to a flexible linker. What's the linker length for Cy3B and the other two protrusions? It is not clear from the figures presented.

The pointer has a flexible linker length of 19 nucleotides in total with the last 7-8 nucleotides being involved in the binding. The protruding docking strands contain the overlap sequence of 7 - 8 nucleotides plus three additional thymine bases extending from

the DNA origami. The oligomers were purchased from Eurofins. This information is given in Supplementary Table SN7.4.

- In the single-color data, should there not be three states? Bound-quenched (state 1), bound-unquenched (state2), and unbound state (not defined in the scheme). A control experiment without the state-1 and state-2 protrusions would reveal the undefined state.

The reviewer is correct that there should be a third, non-bound state. This is true, not only for the single-color data, but for all DNA origami constructs. However, the unbound state is too transient to be observable during the integration time of the TIRF data (≥ 34.2 ms). Below, we show data from MFD-PIE experiments on the two-color, two-state L-shaped DNA origami structure with 8 nt mismatch binding strands (Figure R2, left panel). Here, the intermediate population is observable on the sub-millisecond timescale. When measuring on a DNA origami structure without binding sites, as suggested by the reviewer, the intermediate is clearly observable (Figure R2, right panel). The MFD-PIE data are the subject of a second publication in preparation. As the confocal characterization of sub-millisecond data using MFD-PIE data detracts from the main message of the manuscript, we have not incorporated these details in the current paper.

Figure R2: Two-dimensional single-molecule FRET efficiency versus Stoichiometry histograms recording using multiple-parameter detection with pulsed Interleaved excitation (MFD-PIE). Two-color, two-state DNA-origami structures were measured with 8 nt mis match binding strands (left) and in the absence of binding strands (right).

- Why are the binding times of state-1 and state-2 for single-color data are substantially different given that they are composed of the same sequence? Also, for a 7.5nt complementary strands hybridization, the measured binding time is longer than previously reported

(<https://pubmed.ncbi.nlm.nih.gov/20957983/> and <https://pubmed.ncbi.nlm.nih.gov/32787168/>). Especially that they performed their experiments in low salt buffer (only 50 mM KCl and 100 Tris-HCl (pH?)), the expected binding times are lower than the reported ones. An explanation and discussion on this are required.

We thank the reviewer for this insightful observation. The asymmetry of the binding kinetics has multiple origins. 1) Given the details of the origami design, the unbound tether could not be positioned directly in between the two binding sites. This yields an intrinsic

biased towards one binding site. 2) The position and stress on the two binding strands exciting the origami may not be identical. 3) The dye on the tether, Cy3B, transiently interacts with the surface of the DNA origami near the location of the low-FRET binding position (See the low-FRET tail of Figure R2, right panel). In addition, the acceptor Atto647N can transiently interact with single-stranding DNA, blocking the docking strand. Hence, the fluorophore used and the local DNA environment can have an influence on the kinetics. Transient sticking of Cy3B to DNA origami leads to slower rates compared to previously published measurements using Atto542 (Krause et al., ACS Nano 2021). 4) DNA hybridization kinetics are strongly affected by different external parameters such as temperature (Supplementary Figure SN6.5), salt concentration, etc.).

In addition to the above parameters, it is difficult to compare our values to those published in the references given above. First of all, both publications use much higher salt concentrations > 500 mM NaCl₂ (although we do have 12.5 mM MgCl₂ in our buffer) and they use different DNA sequences. Furthermore, the off-times measured by Jungmann et al and Eklund et al are not comparable to what we measure. Due to the dilute concentration of imager strands, the DNA diffuses away after dissociation. In our DNA origami, the tether can quickly rebind to the same docking strand. Not every dissociation event will lead to a switch to the other binding site. In addition, in the DNA-PAINT constructs, the fluorophores are far from the DNA origami and have less chance to interact with the origami structure. In light of all the factors that influence the rate of DNA hybridization and the differences in the constructs used, the differences in the kinetics rates with published values is not surprising. The rates are in agreement with publications using the same DNA-clock structure (Schroder et al, PNAS 2023). We now mention that the rates we are measuring are different than typically observed in DNA-PAINT. We clarify these issues at the end of the results section for the one-color DNA-origami clock measurements: "We attribute this to an inherent bias in the equilibrium position of the DNA pointer and non-symmetric, non-specific dye-origami interactions. In addition, it is unlikely that the distance to each docking strand and potentially induced stress upon binding are identical for the two binding sites, even though the binding sequence is the same. We note that the kinetics we measure here are not directly comparable to other DNA-hybridization experiments due to both interacting DNA strands are tethered to the origami platform. This leads to a high local concentration of the binder strand and multiple dissociation and rebinding events can occur before the tether switches binding sites."

- A table showing all the binding times for each case is required as it is difficult to read them from figures.

We have added a Supplementary Table with a summary of all the kinetic rates. See. Supplementary Note SN8.

- Authors observe decrease in binding times upon introducing glycerol in the imaging buffer. They explain the difference by taking the hydration shell into account. Another factor that is more interesting in my opinion is to consider the slow down of transitions enabling to clearly observe the individual binding events that were not indistinguishable in low viscous buffer. The fast binding/unbinding which would effectively appear as a

single binding event due to low time resolution is possible because the two strands are right next to each other.

We thank the reviewer for her/his thoughts on this issue. From MFD-PIE measurements, the timescale of transient binding/unbinding in low viscosity buffer is sub-milliseconds. If the unbinding were to be observed, a third state would become visible in the kinetics. Considering the 50 ms integration time of the camera during these measurements, this would be very unlikely.

- Page 3, second paragraph, grammatical error in ‘...for determine of FRET correction..’ should be ‘...for determining FRET correction...’

We thank the reviewer for pointing out the typo. This has been changed.

- ‘...the pretrained neural networks operate locally on each trace, they do not neglect rare events which would be missed in global analysis approaches and lead to inaccurate conclusions.’ It would be great to demonstrate the same for substantiating the claim.

We now substantiate our claim by showing example traces where a global HMM misses transitions that the Deep-LASI analyses catches in Supplementary Figure 4.2.

- Correction in FRET: it is unclear what exactly is corrected.

To convert the measured fluorescence intensity from single donor and acceptor fluorophores into a FRET efficiency, a number of corrections need to be made. These are corrections for spectral crosstalk of the donor fluorescence into the acceptor detection channel, excitation of the acceptor molecule directly rather than via FRET as well as different sensitivities of the instrument to the signal coming from the donor and acceptor fluorophores. That correction factors are needed for accurate FRET efficiency determination is well known in the field. The correction factors, and how they are calculated and implemented to obtain accurate FRET efficiencies are described in detail in Supplementary Note 5.5. We have updated the introduction on page 3 to read: "It classifies each time trace into different categories, identifies which fluorophores are active in each frame, which is then used for determining FRET correction factors for spectral crosstalk, direct acceptor excitation and detection efficiency, and performs a state transition analysis of the different states in dynamic traces."

- In the following sentence, ‘Considering the camera exposure times of 50 ms (6.5 nt, 7 nt and 7.5 nt data sets) and 150 ms (8 nt data set), a dwell-time to frame-time ratio ranges from 10 (6.5 nt State 1) to 62 (8 nt, State 2).’, isn’t the dwell-time to frame ratio 2 (100 msec/50msec) for (6.5 nt State 1) and 22 (3300 msec/150msec) for (8nt state 2)?

We thank the reviewer for bringing this to our attention. We inadvertently provided the rates (which should have been in s^{-1}) rather than dwell times. Also, the frame rate for the 6.5 nt measurements was 34.4 ms. We have now corrected the values. The new text reads: "The summary of all extracted dwell times (Figure 4f, Supplementary Figure

SN6.1) shows an exponential increase in the dwell times of both states with increasing binding site lengths ranging from 0.33 s to 9.5 s. Considering the camera exposure times of 32 ms (6.5 nt), 50 ms (7 nt and 7.5 nt data sets) and 150 ms (8 nt data set) and frame shift time of 2.2 ms, a dwell-time to frame-time ratio ranges from 9 (6.5 nt State 1) to 62 (8 nt, State 2)."

- What's the pH of photostabilization buffer? Surprisingly, the buffer contains low ionic strength without divalent cations. In such buffer, DNA origami structures start disassembling.

We apologize for the misunderstanding. The final buffer used contains 12.5 mM $MgCl_2$ and a starting pH of 8.4. We have updated the Material and Methods to give more details and clearly describe the buffer.

The Methods Sections now reads: "Afterwards, the sample holder was washed 3× with PBS to remove free streptavidin and then with storage buffer (1x TAE, 12.5 mM $MgCl_2$, pH=8.4). Next, the DNA origami sample was diluted to 40 pM in storage buffer, added to the flow chamber and immobilized to the prism surface via the biotin-streptavidin linkage. After 5 min, untethered DNA origami structures were removed by rinsing the chamber 3× with storage buffer. Next, the attached fluorophores on the DNA origami structure were photostabilized by a combination of ROXS and an oxygen scavenging system based on PCA/PCD⁴³. The photostabilization buffer was mixed as follows: 1 μ L of 100 mM Trolox/Ethanol solution was added to 97 μ L storage buffer. The sample was then aged using a UV Lamp (M&S Laborgeräte GmbH, UVAC-6U, 2 × 6 W; 254 and 366 nm) until an equal ratio of Trolox and Trolox-quinone was formed (typically 6 minutes)⁴⁴. Immediately before starting the TIRF experiments, 1 μ L of 100 mM PCA in methanol and 1 μ L of 100 mM PCD solution (50% glycerol, 50 mM KCl, 100 mM Tris HCl, 1 mM EDTA- $Na_2 \times 2H_2O$, pH8) was added to the total volume. In the case of samples containing Atto488, 1 μ L of 14.3 M β ME was added to the photostabilization buffer."

- How long is the enzymatic oxygen depletion time?

We wait 5 minutes after addition of the enzymatic system before starting the measurements and refreshed the buffer every 45 minutes. This is now included in the Material and Methods: "The sample chamber was flushed with photostabilization buffer and sealed to allow for enzymatic oxygen depletion. A minimum of 5 minutes waiting time preceded the experimental recordings. The photostabilization buffer was refreshed every 45 minutes until the end of the experiment."

- Please provide exact mono- and bi-exponential equations used for fitting.

The equations have been added in Supplementary Note SN6.1 and Supplementary Note 6.3.

REVIEWERS' COMMENTS

Reviewer #1 (Remarks to the Author):

The authors have amended the manuscript and satisfactorily addressed my concerns.

Reviewer #2 (Remarks to the Author):

The authors thoroughly worked on the revision addressing all concerns.

Reviewer #3 (Remarks to the Author):

In this new submission, the authors added an important experiment addressing the main concern that was raised about the fairness of the comparison between DeepLASI and HMM trained per-trace. The authors now show that HMMs trained "globally" rather than "locally" perform similarly to Deep-LASI on synthetic data (Figure SN3.6). Further, the authors investigate the difference in performances between global HMM and Deep-LASI and show in Figure SN4.2 that global HMM models struggle to capture fast transitions. As the authors mention this is probably due to heterogeneity in the Markov dynamics within individual traces. This is a very good illustration of the limitations of the current existing methods and thus a great way to illustrate the motivation of the work.

In addition, the authors discuss advantages and pitfalls of their approach in more details. As per the point about interpretability of DNNs that I had raised, I do agree with the authors that training networks on simulated data offers some degree of insight to end-users. As the authors mention, this also provides extensibility to the trained models by re-training or fine-tuning on more extensive simulated data. The authors also mention that DNNs are not interpretable, which is a limitation that must be addressed in the future.

The misunderstanding I had raised were addressed adequately by the authors and the minor issues were fixed.

The HMM is now described sufficiently precisely and to the best of my knowledge, the comparison between HMM and Deep-LASI is fair.

Reviewer #4 (Remarks to the Author):

I am satisfied with the revisions made.